# Random Logit Scaling: Defending Deep Neural Networks Against Black-Box Score-Based Adversarial Example Attacks

**Hamid Dashtbani**                                                    *hmd.dashtbani@sharif.edu*
*Department of Computer Engineering*
*Sharif University of Technology*

**Mehdi Dousti Gandomani**                                             *mahdi.dousti@sharif.edu*
*Department of Computer Engineering*
*Sharif University of Technology*

**AmirMahdi Sadeghzadeh**                                             *sadeghzadeh@sharif.edu*
*Department of Computer Engineering*
*Sharif University of Technology*

**Reviewed on OpenReview:** *https://openreview.net/forum?id=CXafPv4aAG*

## Abstract

Machine learning models are increasingly adapted in various domains. However, adversarial examples pose a significant threat to the reliable deployment of these models. In recent years, some powerful adversarial example attacks have been proposed for the fast and query-efficient generation of adversarial examples, even in black-box scenarios, highlighting the need for scalable, low-cost, and powerful defenses. In this work, we present two contributions to the domain of black-box adversarial example attacks and defenses. First, we propose Random Logit Scaling (RLS), a randomization-based defense against black-box score-based adversarial example attacks. RLS is a plug-and-play, post-processing defense that can be implemented on top of any existing ML model with minimal effort. The idea behind RLS is to confuse an attacker by outputting falsified scores resulting from randomly scaled logits while maintaining the model accuracy. We show that RLS significantly reduces the success rate of state-of-the-art black-box score-based attacks while preserving the accuracy and minimizing confidence score distortion compared to state-of-the-art randomization-based defenses. Second, we introduce a novel adaptive attack against AAA, a SOTA non-randomized black-box defense against black-box score-based attacks that also modifies output logits to confuse attackers, demonstrating its vulnerability against adaptive attacks. Code is avaialble at: https://github.com/HamidDasht/RLS-adversarial-defense

## 1 Introduction

The remarkable performance of deep neural networks has led to a widespread increase in demand for their adoption in various domains ranging from image classification and (Deng et al., 2009), face recognition (Kim et al., 2022) to malware detection (Raff et al., 2018; Chaganti et al., 2022; Deldar & Abadi, 2023). However, deep neural networks continue to be vulnerable to a range of threats, including poisoning (Zhang et al., 2024; Chen et al., 2017b; Xiao et al., 2015), model extraction (Tramèr et al., 2016), and adversarial example attacks (Goodfellow et al., 2015; Madry et al., 2018; Al-Dujaili & O'Reilly, 2020; Ilyas et al., 2018; Andriushchenko et al., 2020; Jiang et al., 2025; Ilyas et al., 2019; Cheng et al., 2020; Zhou et al., 2025; Liu et al., 2019), raising serious concerns regarding their safe deployment. Adversarial examples mislead the models by adding tiny perturbations to an input, causing misclassification by the target model. For example, in a classification task, adversarial examples can cause the label assigned by the model to change to an

incorrect one by applying a carefully crafted perturbation to an input image, while making sure that the new adversarial input is highly similar to the original one to the human eye.

Adversarial examples were initially introduced in a white-box scenario (Goodfellow et al., 2015; Madry et al., 2018) in which the adversary has full access to the model's architecture and parameters. Another scenario is the black-box threat model (Andriushchenko et al., 2020; Ilyas et al., 2018; 2019; Al-Dujaili & O'Reilly, 2020), a more realistic threat model, where the attacker can only observe the model outputs. Even in the restrictive conditions of black-box assumption, many powerful adversarial example attacks have been proposed, such as Andriushchenko et al. (2020); Al-Dujaili & O'Reilly (2020), that can generate adversarial examples with only dozens of queries, a modest cost for any capable adversary.

The dangers of adversarial attacks have led to a plethora of work on defenses against these attacks (Goodfellow et al., 2015; Madry et al., 2018; Singh et al., 2023; Li et al., 2025; Deng & Mu, 2023; Hung-Quang et al., 2024; Zhang et al., 2025; Qin et al., 2021). A significant amount of work has been dedicated to improving model robustness using adversarial training (AT) (Goodfellow et al., 2015; Madry et al., 2018; Li et al., 2022; Rice et al., 2020; Stutz et al., 2020; Singh et al., 2023). However, it has been shown that the improvement in adversarial robustness provided by AT comes with a drop in the model accuracy. Furthermore, the additional overhead of adversarial training makes AT a less practical solution. Ensemble methods (Deng & Mu, 2023; Yang et al., 2021) are another defensive approach against adversarial examples. In this approach, the defender aggregates the predictions of an ensemble of ML models when responding to user queries. A critical disadvantage of ensemble methods is that their maintenance, storage, computation, and deployment costs increase with the number of models in the ensemble.

As another approach to defeat query-based black-box attacks, several proposals have been made to leverage randomization for defense (Hung-Quang et al., 2024; Qin et al., 2021; Aithal & Li, 2022; Lécuyer et al., 2019; He et al., 2019; Ma et al., 2023). Notably, it has been demonstrated that simply adding random noise to the input is effective against black-box score-based attacks (Qin et al., 2021) at the cost of a slight decrease in the model accuracy; this is a common issue with other proposed randomization-based defenses. The methods often achieve better robustness at the cost of degrading the model accuracy.

This work presents two key contributions. First, we propose a plug-and-play, zero-effort, and easily deployable defense against black-box score-based attacks, which preserves model accuracy while significantly reducing the effectiveness of state-of-the-art black-box score-based attacks. Second, we introduce the Pendulum attack, an adaptive attack against AAA Chen et al. (2022), a state-of-the-art (SOTA) non-randomized defense. AAA modifies output logits to reshape the attacker's objective function into a sine-like or linear function with an opposing slope to the original loss, while also optimizing for reduced calibration error.

Our proposed defense, Random Logit Scaling (RLS), confuses black-box attackers by outputting confidence scores resulting from randomly scaled logits. We show that our defense substantially reduces the success rate of SOTA attacks compared to state-of-the-art randomization-based defenses and also AAA by conducting experiments on CIFAR-10 and ImageNet datasets and four image classifiers; notably, our defense improves robustness against Square attack Andriushchenko et al. (2020) by up to 80%. Moreover, we show that, unlike the SOTA randomization-based defenses, our defense maintains accuracy and minimizes distortion to the confidence scores at the same time. RLS allows the deployment of test-time defenses without the additional overhead for training, preserves model accuracy, and therefore eliminates the trade-off between robustness and performance.

The contributions of this work can be summarized as follows:

- We propose RLS, a lightweight, zero-effort defensive method against black-box score-based adversarial example attacks, and provide an analysis of the effect of logit-scaling on black-box score-based optimizations that attackers use, showing how RLS makes it difficult for the attackers to tackle the optimizations and how it can preserve the model accuracy.

- We show that RLS significantly reduces the attack success rate of black-box score-based attacks through experiments conducted on ImageNet and CIFAR-10 datasets, evaluating $\ell_\infty$, $\ell_2$, and also $\ell_0$ sparse attacks. In these experiments, we use six SOTA attacks and compare RLS with AAA and three SOTA randomization-based defenses. RLS shows improved robustness over the baselines while

    maintaining accuracy and disturbing confidence scores less than the baseline randomization-based defenses. Additionally, we use Expectation over Transformations (EOT) as an adaptive attack against the randomization-based defenses, showing that RLS outperforms other randomization-based defenses against the adaptive attacker and significantly increases the costs for considered adaptive attacker scenarios.

- We propose a novel adaptive attack targeting AAA (Chen et al., 2022), a SOTA non-randomized defense against score-based attacks, demonstrating how randomized defenses like RLS can provide greater advantages for defenders.

## 2 Related Work

This section outlines the related work in adversarial example attacks and defense research. We discuss the related work under two broad topics, namely, query-based attacks and defense methods.

### 2.1 Query-based attacks

Query-based adversarial attacks repeatedly query the victim model and adjust the adversarial perturbation based on the model output to craft adversarial examples. These attacks are either score-based (Vo et al., 2024; Andriushchenko et al., 2020; Al-Dujaili & O'Reilly, 2020) or decision-based (Vo et al., 2022; Chen et al., 2020; Brendel et al., 2018). Score-based attacks assume an attacker has access to all or some elements of the probability vector, contrary to the decision-based attacks where only the predicted labels are available. Lacking access to score information, decision-based attacks are often more challenging to conduct, thus increasing the number of queries required to create adversarial examples and raising the cost of the attack. We only focus on defending score-based attacks where powerful adversarial example attacks exist that can generate an adversarial example with only dozens of queries compared to thousands that decision-based attacks require.

Black-box score-based attacks lack access to the model parameters; therefore, they cannot compute input gradient of the loss function. Black-box attackers try to estimate the gradient or use a proxy of the gradient. Chen et al. (2017a) proposed a Zeroth Order Optimization (ZOO) method for gradient estimation using the model outputs and then generate an adversarial image using the estimated gradient. Borrowing from conventional derivative-free optimization research, Natural Evolution Strategies (NES) Ilyas et al. (2018) as another approach to estimate the gradient. Bandit (Ilyas et al., 2019) uses the similarity of gradients from one step to the next in an iterative attack, and the similarity between neighboring pixels. Bandit exploits these correlations as prior information for more efficient gradient estimation. Al-Dujaili & O'Reilly (2020) proposed SignHunter which improves attack efficiency by eliminating the gradient estimation and only relying on the gradient sign. Andriushchenko et al. (2020) proposed a very powerful and query-efficient adversarial attack that uses square-shaped adversarial noise to solve the black-box optimization using random search. Square attack demonstrates a high success rate while requiring only a few dozen queries. BruSLe (Vo et al., 2024) is a SOTA sparse score-based attack that employs a Bayesian framework and a learnable mask to identify sparse perturbations under an $\ell_0$ constraint.

### 2.2 Defense Methods

Numerous efforts have been made to mitigate the effectiveness adversarial example attacks. One category is adversarial training, where DNNs are trained on both clean and adversarial samples (Tramèr et al., 2018; Rice et al., 2020; Stutz et al., 2020; Li et al., 2022). Although adversarial training has been proven effective to some extent, it has several limitations, including its high computational cost, reduced accuracy for benign samples, and the risk of overfitting to adversarial examples. Ensemble methods (Deng & Mu, 2023; Yang et al., 2021) aggregate the prediction of an ensemble of models in their response but increase the cost of storage and computation. Chen et al. (2022) introduce AAA and propose optimizing output logits so that the output loss takes the shape of a target curve (specifically, a sine function or a linear function with a slope opposite to the true loss function's slope) to confound the attacker. We show how the deterministic nature of AAA can be used by attackers to evade this defense by proposing a simple adaptive attack against AAA-sine. Additionally, AAA introduces additional overhead due to its logit optimization process, which can

be problematic in applications that rely on fast and real-time predictions, e.g., self-driving cars speeding on highways.

Randomization-based defenses introduce randomness into the model predictions by randomizing inputs (Qin et al., 2021), outputs (Aithal & Li, 2022), parameters (He et al., 2019; Ma et al., 2023), or the intermediate output features (Hung-Quang et al., 2024) of the model. These methods improve robustness against adversarial attacks, but they can have a negative effect on the model performance by degrading its accuracy. Random Noise Defense (Qin et al., 2021) proposes randomly adding a small Gaussian noise to the inputs. Boundary defense (Aithal & Li, 2022) proposes adding a random noise to logits. Hung-Quang et al. (2024) investigate the effect of adding small random noise to the output of intermediate layers of neural networks and concludes that adding a small noise to the outputs of the penultimate layer improves robustness against adversarial example attacks.

Existing defensive methods involve significant trade-offs, either in terms of computation or the model's accuracy. One of the key challenges tackled by our proposed method is to balance robustness against attacks and its performance. In other words, our lightweight method can defend against score-based attacks without affecting the model's accuracy and performance.

## 3 Preliminaries

This section discusses some crucial preliminary concepts related to adversarial example attacks. Section §3.1 discusses adversarial examples and their various categories, Sections §3.1.1 and §3.1.2 introduce two popular white-box adversarial example attacks, and Sections §3.2.1 and §3.2.2 discuss black-box score-based and decision-based attacks, respectively.

### 3.1 Adversarial Examples

An adversarial example $x_{adv}$ results from carefully perturbing a sample input such as $x$, where $x_{adv}$ leads to an incorrect prediction. In other words, for a model $f_w$, where $w$ denotes the model parameters, an adversarial example $x_{adv}$ can be defined as

$$x_{adv} = x + \delta$$
$$\text{s.t.} \quad \bar{f}_w(x_{adv}) \neq \bar{f}_w(x)$$

where $\bar{f}_w(x)$ denotes the predicted label for input $x$ and $\delta$ represents the carefully crafted perturbation by the attacker.

**Untargeted and Targeted Attacks**: Adversarial example attacks are either untargeted or targeted. For targeted attacks, an adversarial example $x_{adv}$ is created for an original sample $x$ with label $y$, such that $x_{adv}$ is classified as a specific class $y_{trg}$ where $y_{trg} \neq y$. In the untargeted setting, however, the attacker's only goal is that $x_{adv}$ is classified as $y_{adv}$ where $y_{adv}$ can be any class different from the correct one: $\bar{f}_w(x_{adv}) \neq \bar{f}_w(x)$.

### 3.1.1 Fast Gradient Sign Method (FGSM)

FGSM Goodfellow et al. (2015) creates the $x_{adv}$ for a sample $x$ with true label $y$ by moving in the input gradient direction

$$x_{adv} = x + \alpha \cdot \text{sign}(\nabla_x(\ell(f_w(x), y)))$$

where $\ell$ is the loss function (e.g. cross entropy) , $\alpha \geq 0$ controls the magnitude of the step, sign denotes the sign function, and $\nabla_x(\ell(f_w(x), y))$ represents the input gradient of the loss function. In other words, the idea behind FGSM is to use this gradient to craft an adversarial sample $x_{adv}$ that maximizes the loss value for the true label $y$, causing the model $f_w$ to misclassify $x_{adv}$.

### 3.1.2 Projected Gradient Descent

Madry et al. Madry et al. (2018) generalized FGSM by introducing an iterative adversarial example generation method named Projected Gradient Descent (PGD) formulated as

$$x'_{t+1} = \text{PGD}_{x,\epsilon} \left( x'_t + \alpha \cdot \text{sign} \left( \nabla_x \ell(f_w(x'_t), y) \right) \right)$$

where $x'_t$ denotes the adversarial example at iteration $t$, $\text{PGD}_{x,\epsilon}$ denotes the projection function within the $\epsilon$-ball of $x$, it ensures that at each step $||x - x'_t||_p \leq \epsilon$.

One benefit of PGD is that its iterative nature and projection mechanism lead to more refined perturbations and prevent significant changes to the resulting adversarial example by keeping the intermediate samples within the $\epsilon$-ball around the original $x$.

### 3.2 Black-box Adversarial Example Attacks

While PGD and FGSM assume white-box access, in black-box settings the attacker only has an oracle access, meaning the attacker can only observe the output for any query $q$ sent to the model. This lack of access to the parameters prevents black-box attacks from computing the input gradient. However, black-box attacks use alternative methods to overcome this challenge. Black-box attacks are further categorized as either score-based or decision-based attacks, as discussed in §3.2.1 and §3.2.2.

### 3.2.1 Black-Box Score-Based Attacks

Score-based attacks assume access to confidence scores (Guo et al., 2019; Ilyas et al., 2019; Andriushchenko et al., 2020; Ilyas et al., 2018). Here, we discuss two common approaches to score-based attacks.

**Zero-Order Optimization**: An alternative approach to direct gradient computation is to estimate the gradient of the adversary's objective function $\mathcal{J}$ using the output values of model $f$ for an input $x$ and slightly modified versions of $x$ (Ilyas et al., 2019; 2018). This gradient estimator can be formulated as

$$g(x) = \frac{\mathcal{J}(f(x + \mu u)) - \mathcal{J}(f(x))}{\mu} u$$

in which $u$ is drawn from the Normal distribution $\mathcal{N}(0, I)$ and $\mu \geq 0$ determines the size of the perturbation.

**Random Search**: An alternative approach to gradient estimation is using search-based methods (Andriushchenko et al., 2020; Cheng et al., 2020). These methods iteratively sample a candidate direction $u$ from a pre-defined distribution, apply it to the sample $x$, and test the effect of moving the sample in the direction of $u$ by computing $h(x)$ defined as

$$h(x) = \mathcal{J}(f(x + \mu u)) - \mathcal{J}(f(x)) \tag{1}$$

in which $\mu \geq 0$ is step magnitude. At each iteration, search-based methods replace $x$ by $x + \mu u$ if $h(x) \leq 0$, i.e., if $x + \mu u$ causes the prediction score of the correct class to decrease.

### 3.2.2 Black-Box Decision-Based Attacks

Decision-based black-box attacks differ from score-based attacks in that they assume the model $f$ outputs only the final prediction (Brendel et al., 2018; Chen et al., 2020). In other words, it only outputs the label of the top scoring class. Decision-based attacks usually start with a significantly perturbed and misclassified sample $x_{adv}$, then aim to reduce the dissimilarity between $x_{adv}$ and the original $x$ while ensuring that the resulting sample is still misclassified.

## 4 Attacking AAA

Chen et al. (2022) proposed Adversarial Attack on Attackers (AAA) for defense against black-box score-based attacks. AAA optimizes output logits such that the objective loss function computed by attackers

is transformed into a sine-like shape (AAA-sine) or a linear shape with an opposite slope to the true loss. It was shown that AAA-linear is vulnerable to a simple adaptive attack (Chen et al., 2022) (the attacker can simply estimate the gradient and step in the opposite direction or search for the opposite direction in search-based attacks). However, here we propose an effective adaptive attack against AAA-sine to show that AAA is vulnerable to adaptive attackers.

We propose Pendulum attack, an adaptive attack against AAA-sine that repeatedly changes the direction the attacker follows. Both random-search and gradient-based attacks generally follow the direction that minimizes the loss, hence, AAA-sine misleads attackers by reversing the gradient direction in some parts of the loss curve, effectively trapping them in local minima. Our Pendulum adaptive attack works as follows: when the attacker fails to find a direction that reduces the loss for $k$ attack iterations, they shift to the opposite direction, one that increases the loss. After several iterations, however, the attacker may become stuck at a local maximum of the sine-like loss curve. To overcome this, after another $k$ iterations with no success in moving up the loss curve, the attacker again changes direction. By iteratively switching directions upon $k$ unsuccessful consecutive attack iterations, the attacker will be able to climb up the local minima in the loss curve that AAA-sine generates and climb back down from local maxima again when it fails to find an upward direction for $k$ attack iterations. Our results show that the attack success rate of attacks incorporating the adaptive Pendulum attack improves by more than 35% (as detailed in Section 6.2).

## 5  Random Logit Scaling

In this section, we propose Random Logit Scaling (RLS) as a defense against black-box score-based adversarial example attacks. We highlight that since RLS preserves predicted label and only perturbs confidence scores, it is not effective against decision-based attacks.

### 5.1  Objective Functions of Black-Box Score-Based Attacks

The form of the adversary's objective function $\mathcal{J}$ varies depending on whether the attack is targeted or untargeted. In the targeted adversarial example attacks, the attacker aims at generating a perturbed sample $x' = x + \delta$ from an original sample $x$ with true label $y$ such that $x'$ is classified as $t$ where $t \neq y$. The optimization in the targeted scenario can be formulated as

$$\operatorname*{argmin}_{\delta} \max_{k \neq t} f_k(x + \delta) - f_t(x + \delta) \tag{2}$$

where $f_i(x)$ denotes the score of class $i$ for input $x$.

In the untargeted setting, the attacker only wishes to generate a perturbed sample $x'$ such that it is not classified as the true class $y$. In other words, the attacker only aims for $x'$ to be misclassified as any class other than $y$

$$\operatorname*{argmin}_{\delta} f_y(x + \delta) - \max_{k \neq y} f_k(x + \delta) \tag{3}$$

In both classes of score-based attacks discussed in Section 3.2.1, namely search-based and gradient estimation-based attacks, the attacker essentially modifies output vectors $f(x')$ by applying carefully crafted perturbations to $x'$ and using

$$\mathcal{J}(x') = \max_{k \neq t} f_k(x') - f_t(x') \tag{4}$$

for targeted attacks and

$$\mathcal{J}(x') = f_y(x') - \max_{k \neq y} f_k(x') \tag{5}$$

for untargeted attacks to guide the minimization objectives in Eq. (2) and Eq. (3), respectively.

We highlight that the objective functions in Eq. (2) to (5) are representative margin loss objectives rather than an exhaustive description of all score-based attacks. Generally speaking, score-based attacks may use

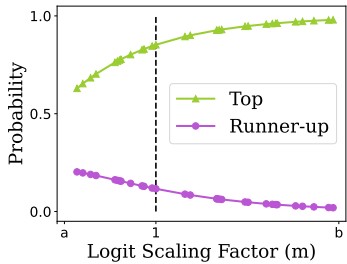

Figure 1: The effect of scaling logits by a factor of $m$ on the probabilities of the top two classes.

surrogate losses or alternative optimization procedures that essentially pursue the same goal. However, RLS generalizes across score-based attacks because it corrupts the score information on which both exact and surrogate objectives rely.

## 5.2 The Method Overview

The intuition behind our proposed defense is that by modifying $f(x')$ in a deceptive way, the attacker will not be able to tackle either of the optimizations Eq. (2) or Eq. (3). More concretely, a defender could *blind* an attacker by not producing the true logit values, resulting in falsified confidence scores. Outputting a greater or smaller logit value for each class $i$ leads to a falsified score $f_i(x)$ and a false value for the objective functions Eq. (2) and Eq. (3). In other words, masking the scores in a way that falsifies the gaps in Eq. (4) and Eq. (5) undermines an attacker's effort to minimize the objective values. However, this masking should be done with care since arbitrary modification of logits could lower the model accuracy as the value of each logit directly reflects the ranking of the respective class. A possible approach is to scale output logits by a positive factor $m$. Scaling logits with a positive value can falsify the values in Eq. (4) and Eq. (5) while preserving the ranking of output classes as explained in Sections 5.2.1 and Appendix B.

### 5.2.1 The Effect of Logit Scaling

Using different values of $m$ to scale the output logits, results in large fluctuations in the value of the objective functions used by the attacker, hence confusing the attacker and making it difficult for both gradient-based and search-based black-box attacks to tackle the optimization problem:

**Untargeted Attacks Eq. (5)**: As long as the untargeted attack is unsuccessful (as long as Eq. (5) is non-negative), $\max_{k \neq y} f_k(x')$ represents the runner-up class score, and the top-scoring class is the true class ($f_y(x')$). Hence, Eq. (5) effectively subtracts the runner-up class score from the top-scoring class score, i.e., it uses the gap between the two classes to guide the optimization in Eq. (3). However, as the analysis in Appendix B and Appendix C show, scaling the logits with the random coefficient $m$ can cause a falsified lower or higher value for the scores and the gap between the two classes used in Eq. (5).

**Targeted Attacks Eq. (4)**: Scaling logits by a positive $m$ affects Eq. (4) similarly to the untargeted attacks. As long as the targeted attack is not successful, $\max_{k \neq t} f_k(x')$ represents the top-scoring class, and the target class score $f_t(x)$ is a lower-ranked class. As we show in Appendix B, the gap between any two classes is affected by the fluctuations caused by logit scaling in a similar manner.

We investigate the score fluctuations of any two classes in Appendix B and also the fluctuations to the top-scoring class in Appendix C. As a visual example, Figure 1 demonstrates how scaling the logits with different values of $m$ affects the probability scores of the two top-ranking classes. Here, both $a$ and $b$ are positive values with $a < b$. As $m$ moves toward $b$, the gap between the two classes increases. Generally, when a larger $m$ is selected, i.e., when $m > 1$, the absolute value of all the class logits is increased. As a result, the confidence score of the true class increases. On the other hand, when $m$ approaches zero, the probability vector will get smoothed out, which causes the probability of the top-scoring class to decrease while the lower-scoring classes will generally have higher confidence scores but still lower than the original top-ranked class.

### 5.2.2 Random Logit Scaling

We propose Random Logit Scaling (RLS) to blind the attacker and mask the confidence scores while maintaining the model accuracy. RLS scales the logits in order to cause a false increase or decrease in the objective values utilized by an attacker, as explained in Section 5.2.1. RLS scales logits $z$ by a coefficient $m$ sampled from a continuous uniform distribution $\mathcal{U}$ with a range $(a, b)$ where $a > 0$ and $b > 0$

$$z' = m \times z \tag{6}$$
$$\text{s.t.} \quad m \sim \mathcal{U}(a, b) \tag{7}$$

We found uniform distribution to work surprisingly well, outperforming other randomization-based defenses in means of reducing the attack success rate, preserving the model accuracy, and minimizing distortion to the confidence score. However, we present a brief ablation study on the selection of the probability distribution for RLS and its parameter values in Appendix D.

**RLS Effect on Gradient Estimation and Random Search**: The variations to the confidence scores, caused by sampling a different $m$ each time the model is queried, mask the true value of $f(x')$. Therefore, when the attacker attempts to estimate the gradient of $f(x')$, the computations will be based on falsified scores, resulting in a falsified estimate of the true gradient. Similarly, when the attacker uses search-based methods, such as random search used by Square (Andriushchenko et al., 2020) attack, a query with a small (large) $m$ outputs a fabricated (high) low score for $h(x)$, misleading the attacker to select (skip) the query for the next stage of the attack, as explained in Section 3.2.1.

### 5.3 Threat Model

We consider a black-box scenario in which the model is deployed as a service and users have only oracle access to the model. Specifically, the model returns the scaled logits produced by the defense. We assume that the adversary is aware of the deployed defense and its implementation details; however, the scaling factor m is not known. The attacker is allowed to query the model adaptively and repeatedly on the same input and is subject to no API restrictions other than a query budget of 10,000 queries, which is used throughout all of our evaluations.

## 6 Experiments

In this section, we evaluate and compare the performance and robustness of RLS against various SOTA attacks and defenses.

### 6.1 Experimental Setup

Six SOTA attacks are used in the experiments, namely NES (Ilyas et al., 2018), Bandit (Ilyas et al., 2019), Square (Andriushchenko et al., 2020), SignHunter (Al-Dujaili & O'Reilly, 2020), and ZO-signSGD (Liu et al., 2019). Additionally, we use BruSLe attack (Vo et al., 2024) for evaluations in $\ell_0$ space (as detailed in Appendix F.2). The exact settings and parameter values for each attack are outlined in Appendix A. We compare the results of our defense with that of Randomized Featured Defense (Hung-Quang et al., 2024) (RFD), Random Noise Defense (iRND) (Qin et al., 2021), Boundary Defense (oRND) (Aithal & Li, 2022), and AAA. The exact settings for all the defense methods is outlined in Appendix A.

Our experiments use CIFAR-10 (Krizhevsky, 2012) and ImageNet (Deng et al., 2009) as benchmark datasets. For the experiments on ImageNet, we use Pytorch Torchvision's ResNet-50 with its pre-trained weights. For CIFAR-10, we use VGG-16, ResNet-18, and WideResNet-28-10. We train them and achieve accuracy scores of 91.80%, 95.53%, and 96.20%, respectively. We use a randomly selected set of 1,000 test samples for conducting experiments on CIFAR-10 (unless specified otherwise). For ImageNet we use 1,000 samples from the validation set (same set of samples as in iRND (Qin et al., 2021)). We conduct our experiments over $\ell_\infty$, $\ell_2$, and $\ell_0$ norm spaces. For $\ell_\infty$ we use a perturbation budget of $\epsilon = 0.05$ ($\epsilon = 0.1$ for evaluations in Appendix F.3). Settings for $\ell_2$ and $\ell_0$ attacks are respectively detailed in Appendices F.1 and F.2

## 6.2 Results of the Pendulum Adaptive Attack Against AAA

Following Chen et al. (2022) that evaluated their adaptive attack against AAA-linear by incorporating it into Square attack Andriushchenko et al. (2020), we incorporate the Pendulum attack into Square to adaptively attack AAA-sine.

We report the attack success rate and the average number of queries of successfully crafted adversarial examples for our adaptive attack at values of $k = 5$ and $k = 10$ against AAA-sine in Table 1. Our adaptive attack is referred to as Pen-Square-$k$. The evaluations use ResNet-18 and the randomly selected 1,000 test samples from CIFAR-10. We use the $\ell_\infty$ with a perturbation budget of $\epsilon = 0.05$.

Table 1: Pen-Square performance against AAA-sine.

| Attack | ASR | AVG Queries |
|---|---|---|
| Square | 45.80% | 114.51 |
| Pen-Square-5 | 74.88% (+29.08%) | 2369.62 |
| Pen-Square-10 | 81.42% (+35.62%) | 2946.09 |

The results highlight the vulnerability of AAA to our proposed adaptive attack. Nonetheless, we provide a comparison between RLS and AAA in Appendix F.4 demonstrating that RLS outperforms AAA even in scenarios where no adaptive attack is applied to AAA.

## 6.3 Confidence Score Distortion and Calibration Error

Randomized-defenses cause distortion to model outputs and scores. Scaling logits distorts the confidence scores too, which can be harmful, especially in applications where the magnitude of the prediction's confidence score is crucial—scaling logits up results in a misleadingly higher confidence score for the top-scoring class while shrinking logits results in a lower score for the class (see Appendix C). This can pose a challenge in applications that rely on the accuracy of the gap between the model's top and runner-up classes' confidence scores (for instance, in out-of-distribution detection (Lee et al., 2018), out-of-distribution samples typically exhibit a smaller gap and in medical applications Liang et al. (2018), it is essential to rely on outputs with high confidence for decision-making). We test the degree of confidence score distortion in the presence of our defense and compare our results to Randomized Feature Defense (RFD) (Hung-Quang et al., 2024), Random Noise Defense (Qin et al., 2021) (iRND), and boundary defense (Aithal & Li, 2022) (oRND).

In contrast to other randomized defenses, our method keeps the predicted class unchanged. This means that the model accuracy does not decrease in the presence of RLS. On the other hand, iRND, RFD, and oRND can cause the defended model to misclassify some input samples, hence lowering the model accuracy.

Table 2 reports the distortion of true probability values in the presence of our defense, iRND, RFD, and oRND. We used all the 10,000 test samples of CIFAR-10 to conduct this experiment. We first collect the original probability vectors for all the samples for an undefended ResNet-18 model. Then, we implement each defense and collect the probability vectors for the defended model. Finally, we compute the $L_2$ norm of the difference between the two vectors for each data sample and report the maximum, average, and minimum taken over all the differences vectors.

The results show that RLS causes a lower level of distortion on average compared to iRND, oRND, and RFD for $m \in (0.5, 10)$, and even when sampling from a large range of possible coefficients, such as $(0.5, 1000)$, RLS outperforms both iRND and RFD but distorts the score slightly more than oRND. The maximum distortion that our method causes is also lower than that of other defenses since other defenses can cause a sample to be misclassified.

Additionally, in Table 3, we evaluate the expected calibration error of RLS and compare it with other defensive methods on ResNet-18. Our evaluations show that RLS (0.5, 10) and RLS (0.5, 1000) cause an average of 1.3% and 1.8% increase in the ECE, respectively. This means RLS increases ECE more than RFD, AAA, and oRND but less than iRND.

Due to the confidence score distortion introduced by RLS, it creates a trade-off between robustness and confidence score preservation in applications that heavily rely on accurate confidence estimates. RLS is

Table 2: Confidence score distortion in the presence of RLS and other randomized defenses for ResNet-18 over all the 10,000 test samples of CIFAR-10.

| Defense | Parameters | Score Differences | | |
|---|---|---|---|---|
| | | Min | Avg | Max |
| iRND | $v = 0.01$ | **1.2e-6** | 0.033 | 1.354 |
| | $v = 0.02$ | 4.5e-6 | 0.087 | 1.413 |
| oRND | $c = 1$ | 7.5e-6 | 0.024 | 1.075 |
| RFD | $\sigma = 2.5$ | 9.3e-6 | 0.028 | 1.19 |
| RLS | $(0.5, 10)$ | 2.3e-5 | **0.022** | **0.68** |
| | $(0.5, 100)$ | 2.3e-5 | 0.026 | 0.791 |
| | $(0.5, 1000)$ | 2.3e-5 | 0.027 | 0.791 |

Table 3: Minimum, average, and maximum ECE of ResNet-18 for each defense under the 10,000 test samples of CIFAR-10. Five executions are used for taking the average and the bin size is 100.

| Defense | Min | Avg | Max |
|---|---|---|---|
| No Defense | 0.0268 | - | - |
| AAA-sine | 0.0220 | 0.0220 | 0.0220 |
| RLS (0.5, 10) | 0.0331 | 0.0401 | 0.0429 |
| RLS (0.5, 100) | 0.0447 | 0.0448 | 0.0448 |
| RLS (0.5, 1000) | 0.0447 | 0.0448 | 0.0448 |
| iRND ($\nu = 0.01$) | 0.0284 | 0.0302 | 0.0319 |
| iRND ($\nu = 0.02$) | 0.0439 | 0.0454 | 0.0478 |
| RFD ($\sigma = 2.5$) | 0.0261 | 0.0267 | 0.0279 |
| oRND ($c = 1$) | 0.0257 | 0.0263 | 0.0282 |

Table 4: Defense performance on CIFAR-10. A random set of 1,000 samples from CIFAR-10 are used. The average number of queries for successfully crafted adversarial samples and the attack success rates are reported. Bold and underlined texts represent the best and the second-best results, respectively. The perturbation budget is $\epsilon = 12.75/255$ (or 0.05) and the $\ell_\infty$ norm is used.

| Model | Defensive Method | Acc. Drop | Attacks | | | | |
|---|---|---|---|---|---|---|---|
| | | | NES | Bandit | Square | SignHunter | ZO-signSGD |
| ResNet-18 | None | - | 99.48% / 346.56 | 100% / 213.42 | 100% / 115.19 | 100% / 213.15 | 94.56% / 370.82 |
| | iRND ($\nu = 0.01$) | -0.78% | 96.31% / 527.95 | 46.64% / 358.37 | 89.84% / 309.75 | 73.72% / 637.17 | 88.14% / 590.79 |
| | ($\nu = 0.02$) | -3.44% | 88.42% / 927.46 | 54.92% / 463.32 | 79.02% / 538.21 | 60.27% / 859.42 | 75.54% / 841.50 |
| | oRND ($c = 1$) | -0.44% | 87.41% / 1197.85 | 51.1% / 400.41 | 66.19% / 500.10 | 43.12% / 869.9 | 79.61% / 1056.21 |
| | RFD ($\sigma = 2.5$) | -0.64% | 84.90% / 1272.76 | 57.48% / 675.53 | 70.94% / 667.68 | 45.12% / 1306.31 | 76.70% / 1105.04 |
| | RLS (0.5, 10) | **0%** | 79.36% / 1230.0 | 25.73% / 27.19 | 59.85% / 735.44 | 18.83% / 1304.77 | **64.22%** / 1024.36 |
| | (0.5, 1000) | **0%** | **73.03%** / 1388.61 | **1.56%** / 4.4 | **43.05%** / 92.10 | **17.85%** / 978.58 | 65.36% / 1205.99 |
| VGG-16 | None | - | 99.03% / 547.8 | 100% / 337.83 | 100% / 261.01 | 100% / 274.01 | 85.33% / 497.35 |
| | iRND ($\nu = 0.01$) | -0.11% | 97.52% / 772.44 | 27.94% / 386.80 | 69.42% / 750.2 | 84.69% / 522.47 | 76.25% / 915.84 |
| | ($\nu = 0.02$) | -0.51% | 88.8% / 1239.21 | 32.39% / 445.54 | 55.06% / 534.31 | 66.31% / 624.09 | 68.60% / 1168.02 |
| | oRND ($c = 1$) | -0.35% | 79.68% / 1608.92 | 24.98% / 633.23 | 95.48% / 335.14 | 43.5% / 413.28 | 63.5% / 1457.85 |
| | RFD ($\sigma = 0.35$) | -2.99% | 75.31% / 1779.04 | 40.83% / 721.35 | 54.35% / 752.92 | 51.69% / 1136.16 | 55.63% / 1339.44 |
| | RLS (0.5, 10) | **0%** | 55.13% / 1478.32 | 13.27% / 63.34 | 34.42% / 1820.21 | 21.26% / 332.47 | 48.12% / 1243.98 |
| | (0.5, 1000) | **0%** | **47.1%** / 1812.13 | **2.7%** / 7.56 | **21.56%** / 38.58 | **15.85%** / 224.44 | **43.05%** / 1343.00 |
| WRN28-10 | None | - | 100% / 394.72 | 100% / 209.85 | 100% / 119.01 | 100% / 281.72 | 97.72% / 395.94 |
| | iRND ($\nu = 0.01$) | -1.06% | 100% / 551.44 | 49.53% / 339.44 | 81.59% / 603.18 | 66.28% / 895.15 | 95.9% / 607.92 |
| | ($\nu = 0.02$) | -3.95% | **78.31%** / 930.28 | 59.05% / 393.50 | 76% / 635.48 | 57.16% / 1004.21 | 88.51% / 906.94 |
| | oRND ($c = 1$) | -0.32% | 91.98% / 1059.45 | 52.82% / 215.55 | 68.98% / 496.45 | 39.54% / 789.70 | 85.39% / 903.08 |
| | RFD ($\sigma = 0.7$) | -0.72% | 94.60% / 906.26 | 46.16% / 490.04 | 71.75% / 407.27 | 42.67% / 948.99 | 89.03% / 837.22 |
| | RLS (0.5, 10) | **0%** | 79.63% / 1261.88 | 27.24% / 22.85 | 58.74% / 1303.52 | 20.48% / 817.16 | 72.46% / 1058.89 |
| | (0.5, 1000) | **0%** | 79.63% / 1356.57 | **1.15%** / 3.54 | **41.38%** / 127.47 | **15.08%** / 1168.73 | **28.27%** / 1237.24 |

therefore most appropriate in settings where preserving top-1 predictions and improving robustness are prioritized, whereas applications that depend heavily on well-calibrated confidence scores may require additional calibration techniques or a carefully selected distribution for the scaling factor.

### 6.4 Results of Defense against Score-based Black-box Attacks

In this section, we evaluate the resistance of RLS against five state-of-the-art black-box score-based attacks. We compare our results with Random Noise Defense (Qin et al., 2021) (iRND), Randomized Feature Defense (RFD) (Hung-Quang et al., 2024), and Boundary defense (Aithal & Li, 2022) (oRND). We report the performance of RLS on CIFAR-10 and ImageNet in Tables 4 and 5, respectively. Additional results for attacks in the $\ell_2$ norm space, SOTA sparse attack in the $\ell_0$ space (Vo et al., 2024), evaluation for a larger perturbation budget of $\epsilon = 0.1$ in the $\ell_\infty$ space, and comparison with AAA are respectively available in Appendices F.1, F.2, F.3, and F.4. Additional results for attack success across different query budgets of 1000, 2000, 5000, and 10000 is also avaialble in Appendix F.6.

As reported in Table 4, RLS drastically reduces the attack success rate in almost all cases while maintaining the model accuracy. For NES (Ilyas et al., 2018), RLS outperforms SOTA randomization-based methods by at least

Table 5: Defense performance on ImageNet. One thousand samples from the validation set are randomly selected to attack Torchvision's pre-trained ResNet-50. The average number of queries for successfully crafted adversarial samples and the attack success rate are reported. Bold and underlined texts represent the best and the second-best results, respectively. The perturbation budget is $\epsilon = 12.75/255$ (or 0.05) and the $\ell_\infty$ norm is used.

| Model | Defensive Method | Acc. Drop | Attacks | | | | |
|---|---|---|---|---|---|---|---|
| | | | NES | Bandit | Square | SignHunter | ZO-signSGD |
| ResNet-50 | None | - | 99.34% / 1406.45 | 99.6% / 427.95 | 100% / 74.16 | 100% / 314.1 | 79.18% / 1615.24 |
| | iRND ($\nu = 0.01$) | -0.9% | 97.02% / 2091.62 | 53.18% / 598.43 | 85.89% / 133.17 | 64.46% / 401.20 | 77.03% / 1787.08 |
| | ($\nu = 0.02$) | -2.1% | 83.75% / 2998.02 | 53.34% / 579.69 | 75.45% / 179.30 | 52.18% / 505.66 | 67.58% / 2442.84 |
| | oRND ($c = 1$) | -2.2% | 78.42% / 2535.13 | 82.29% / 308.79 | 87.78% / 216.54 | 76.45% / 418.36 | 71.45% / 1876.61 |
| | RFD ($\sigma = 3$) | -0.2% | 79.63% / 2480.08 | 67.53% / 463.93 | 77.52% / 509.46 | 57.97% / 527.00 | 68.27% / 2129.13 |
| | RLS $(0.5, 10)$ | **0%** | 57.82% / 2571.82 | 54.21% / 432.97 | 59.55% / 187.69 | 24.47% / 84.39 | 51.81% / 2134.05 |
| | $(0.5, 1000)$ | **0%** | **56.88%** / 2651.26 | **27.54%** / 21.34 | **48.52%** / 13.76 | **21.41%** / 104.46 | **48.47%** / 2100.21 |

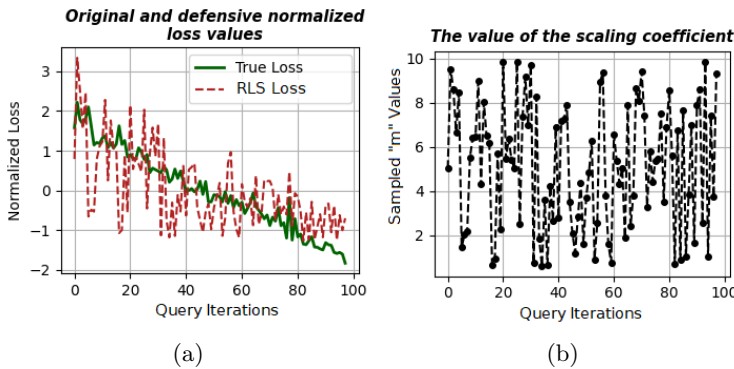

(a)                                         (b)

Figure 2: Objective function loss values and $m$ values for 100 iterations of Square attack for a random CIFAR-10 image in the presence of RLS.

10%, except for WideResNet-28-10, where iRND works slightly better. However, iRND decreases the model accuracy by 3.95%, while RLS preserves the accuracy. For Bandit (Ilyas et al., 2019), Square (Andriushchenko et al., 2020), SignHunter (Al-Dujaili & O'Reilly, 2020), and ZO-signSGD (Liu et al., 2019), RLS consistently outperforms iRND, RFD, and oRND by at least 10% and up to more than 90%. Notably, RLS reduces the attack success rate for the powerful Square attack to only around 21% on VGG-16 and around 40% for both WideResNet-28-10 and ResNet-18.

We also evaluated our defense on a pre-trained ResNet-50 on ImageNet. The results are reported in Table 5. The results reveal that RLS achieves the best and the second-best performance with respect to attack success rate in all the cases except for the Bandit attack, where iRND achieved the second-best performance but at the cost of a 0.9% drop in the model accuracy.

Figure 2 visualizes the effect of RLS on an attacker and how our defense causes the objective function values computed by an attacker to fluctuate. Figure 2a illustrates the normalized loss values for a randomly selected CIFAR-10 image in a few dozen iterations of Square attack. The dashed plot indicates the normalized loss values in the presence of our defense, whereas the solid one plots the values for an undefended model. The plot in Figure 2b, indicates the sampled $m$ value for each query iteration.

## 6.5 Adaptive Attack against RLS

To evaluate the robustness of our defense against adaptive attacks, we use Expectation over Transformation (EOT) (Athalye et al., 2018) attack. EOT uses $k$ queries for each sample at each attack iteration and takes an average over the outputs to cancel out the randomness introduced by randomized defenses. We report our results in Table 6 for $k = 1$, $k = 5$, and $k = 10$ while additional results for $k = 25$, $k = 50$, and $k = 100$ are presented in Appendix F.5. We use Square attack (Andriushchenko et al., 2020) and limit the number of

attack iterations to 1,000 (5,000 queries for $k = 5$ and 10,000 queries for $k = 10$) and use a randomly selected set of 2,000 test samples from CIFAR-10 to attack our trained ResNet-18 model. As the results show, our method can withstand the adaptive attack better. While the adaptive attack can achieve almost 80% success rate against iRND and 70% against oRND and RFD, our method can still decrease the attacker's success rate by at least around 45%.

The results also show that when the defender uses a wider range of $m$ values, the attacker's success rate decreases. This is because as the range of possible $m$ values increases, the attacker has to use more queries when taking the average to estimate the true confidence scores. In other words, the attacker has to increase $k$ when the defender uses a wider range of $m$ values. In Appendix E, we have plotted correlation between true loss and loss under RLS for $k = 5, 10, 20$, and 100.

Table 6: Defense robustness against the EOT adaptive attack incorporated into the Square attack on ResNet-18 using 2,000 samples from CIFAR-10. The perturbation budget is $\epsilon = 12.75/255$ (or 0.05) and the $\ell_\infty$ norm is used.

| Defense | Parameters | $k = 1$ | | $k = 5$ | | $k = 10$ | |
|---------|-----------|---------|---------|---------|---------|----------|---------|
| | | ASR | Avg. Q. | ASR | Avg. Q. | ASR | Avg. Q. |
| | $(0.5, 10)$ | 40.48% | 46.70 | 50.50% | 1287.45 | 55.48% | 5164.02 |
| RLS | $(0.5, 100)$ | **35.97**% | 11.59 | 42.79% | 208.16 | 44.94% | 1348.93 |
| | $(0.5, 1000)$ | 36.34% | 10.37 | **41.22**% | 101.20 | **42.53**% | 362.07 |
| iRND | $\nu = 0.01$ | 56.76% | 47.60 | 77.36% | 682.75 | 82.70% | 1828.56 |
| | $\nu = 0.02$ | 57.01% | 59.40 | 67.19% | 815.07 | 74.17% | 2458.26 |
| oRND | $c = 1$ | 56.51% | 55.74 | 66.41% | 662.26 | 69.91% | 2380.21 |
| RFD | $\sigma = 2.5$ | 58.90% | 48.63 | 66.81% | 863.52 | 71.59% | 3539.17 |

## 7 Discussion

Here we discuss the limitations of RLS. First of all, RLS preservers the predicted class and primarily only perturbs the confidence scores, therefore it has limited effectiveness against decision-based attacks that only rely on the model's final prediction. Nevertheless, existing state-of-the-art decision-based attacks require substantially more queries than score-based attacks. Overall, RLS should be viewed as a defense mainly against score-based adversaries rather than a comprehensive defense against all black-box attacks.

Secondly, RLS is inherently a randomized defense and provides no certified robustness guarantees. Therefore, a sufficiently capable attacker with a large enough number of repeated observations for the same input can potentially eliminate the effect of the randomness. However, this would substantially increase the number of required queries and the attack cost, while making the detection of adversarial attempts easier.

## 8 Conclusion

In this work, we proposed Random Logit Scaling (RLS), a randomization-based, lightweight, and powerful defense against black-box score-based adversarial examples attacks and proposed an effective adaptive attack against AAA. Through extensive experiments on two benchmark datasets, six SOTA attacks, four state-of-the-art defenses, and four image classifiers, we showed that RLS preserves the model accuracy, minimizes distortion to the confidence scores, and outperforms other defenses against black-box score-based attacks. Furthermore, we showed that RLS is more robust against the EOT adaptive attack. We also analyzed how scaling logits affects a model and black-box adversaries and how it can be an effective defense against adversarial example attacks. The significant resistance of RLS against the most powerful black-box attacks, combined with its zero-effort implementation and preservation of the model accuracy, makes it a very powerful defense against black-box adversarial attacks.

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

# A Attack and Defense Configurations

Table 7: Hyperparameter values for attacks. Both $\ell_\infty$ and $\ell_2$ spaces use the same hyperparameter values.

| Attack | Hyperparameter | CIFAR-10 | ImageNet |
|--------|----------------|----------|----------|
| NES | Learning Rate ($\eta$) | 0.01 | 0.005 |
| | # of samples for estimation per step ($q$) | 30 | 60 |
| Bandit | Learning Rate ($\eta$) | 0.01 | 0.01 |
| | Learning Rate for Online Convex Optimization ($h$) | 0.1 | 0.0001 |
| | Bandit Exploration ($\delta$) | 0.1 | 0.1 |
| | Tile Size (Data Prior) | 20 | 50 |
| Square | initial precentage of pixels to change ($p$) | 0.05 | 0.05 |
| SignHunter | Step Size ($\eta$) | 0.05 | 0.05 |
| ZO-signSGD | Learning Rate ($\eta$) | 0.01 | 0.005 |
| | # of samples for estimation per step ($q$) | 30 | 60 |
| BruSLe | $m_1$ | 0.24 | 0.24 |
| | $m_2$ | 0.997 | 0.997 |
| | $\lambda_0$ | 0.3 | 0.3 |
| | $\alpha^{\text{prior}}$ | 1 | 1 |

For the Square (Andriushchenko et al., 2020), Bandit (Ilyas et al., 2019), NES (Ilyas et al., 2018), ZO-signSGD (Liu et al., 2019), and SignHunter (Al-Dujaili & O'Reilly, 2020) attacks, we use the implementations provided by Zheng et al. (2023) in all of our experiments. The exact values of hyperparameters for all of the attacks are reported in Table 7. Hyperparameter values for BruSLiAttack (Vo et al., 2024) are also outlined in the table.

For the defensive methods used in all of our experiments, we use the configurations as outlined in the following. For iRND (Qin et al., 2021) we use the two original configurations of $\nu = 0.01$ and $\nu = 0.02$. For oRND (Aithal & Li, 2022) we use $c = 1$. For AAA (Chen et al., 2022) we use the configurations provided by the official implementation, specifically, we use 4 for attractor interval, 5 for regularization weight, learning rate of 0.1, and 100 iterations for optimization. However, authors of RFD (Hung-Quang et al., 2024) do not report the specific hyper-parameter values ($\sigma$) but instead, report the average accuracy drop of the configurations they use. Interestingly, this is because the $\sigma$ value used by RFD has to be tuned for each model specifically. We conducted a search for each of our models and selected the best performing $\sigma$ that provided a higher robustness against the majority of the attacks. We use $\sigma = 0.35$, $\sigma = 2.5$, $\sigma = 0.7$, and $\sigma = 3.0$ respectively for VGG-16, ResNet-18, WideResNet-28-10, and ResNet-50.

## B  The RLS Formulation

We provide a formulation of RLS and how it affects the vector of confidence scores and the attacker's efforts to generate adversarial examples. Specifically, we show how different values of $m$ causes the gap between any two class scores to fluctuate, hence having a direct impact on the values in Eq. (4) and Eq. (5) of Section 5.1.

**Notation**. For any arbitrary input sample such as $x$, $Z$ denotes the original logit vector and $z_i \in Z$ represents the vector's $i^{\text{th}}$ element. $m \sim \mathcal{D}$ represents a randomly selected coefficient $m$ from the distribution $\mathcal{D}$. $p_i \in P$ represents the true probability vector's $i^{\text{th}}$ element, while $p_i' \in P'$ represent the $i^{\text{th}}$ element of the probability vector resulting from multiplying $Z$ by $m$. Finally, $S$ and $S'$ represent the normalization factor of the softmax function for the true and scaled logits, respectively.

Assuming $p_i \in P$ and $p_j \in P$ are the probability scores of two randomly selected classes $i$ and $j$, we have

$$p_i = \frac{e^{z_i}}{S}, \qquad p_j = \frac{e^{z_j}}{S} \tag{8}$$

assuming $i$ is the class with a higher confidence score, since $0 < p_i < 1$ for any $i$, we can write

$$\frac{p_i}{p_j} = \frac{e^{z_i}}{e^{z_j}} > 1 \tag{9}$$

After multiplying the logits by $m$, the resulting class probabilities are

$$p_i' = \frac{e^{mz_i}}{S'}, \qquad p_j' = \frac{e^{mz_j}}{S'} \tag{10}$$

by dividing the probabilities in the same manner, we have

$$\frac{p_i'}{p_j'} = \frac{e^{mz_i}}{e^{mz_j}} = \left(\frac{e^{z_i}}{e^{z_j}}\right)^m \tag{11}$$

by combining Eq. (9) and Eq. (11), we have

$$\frac{p_i'}{p_j'} = \left(\frac{p_i}{p_j}\right)^m \tag{12}$$

based on Eq. (9) and Eq. (12), we can conclude that for $m > 0$

$$\frac{p_i'}{p_j'} > 1 \tag{13}$$

$$p_i' > p_j' \tag{14}$$

therefore, the ranking of the classes is preserved. Additionally,

1. for $0 < m < 1$, the value of $\frac{p'_i}{p'_j}$ decreases towards one

2. for $m > 1$ the value of $\frac{p'_i}{p'_j}$ gets larger

3. for $m < 0$, the value of $\frac{p'_i}{p'_j}$ moves below 1, which means $p'_j > p'_i$, hence the accuracy can drop

Since our method only considers positive values for $m$, we only focus on the first two cases. As $m$ grows larger, $p'_i/p'_j$ moves toward infinity, intuitively, causing $p'_i$ and $p'_j$ to get farther away from each other and resulting in the gap between the probabilities having a higher value than the true one. When $m$ is below one and approaches zero, $p'_i/p'_j$ decreases towards one. This means the values of the manipulated probabilities get closer to each other. In Appendix C, we prove that in both cases, the changes to the gap affect the top-class probability, hampering the effectiveness of many of the gradient estimation and search-based methods, as explained in Sections 5.2.1 and 5.2.2.

## C   The Proof of Fluctuations to The Top-Scoring Class

We assume $t$ represents the top class for a classification task with $n$ classes. We use the notations from Appendix B. First, we have

$$e^{z_t} > e^{z_i}, \quad \forall \quad i \neq t \tag{15}$$

since $m > 0$ and $e^x > 0$ for any $x$, we can derive

$$e^{mz_t} > e^{mz_i}, \quad \forall \quad i \neq t \tag{16}$$

$S$ and $S'$ are defined as follows

$$S = e^{z_1} + \cdots + e^{z_n} \tag{17}$$
$$S' = e^{mz_1} + \cdots + e^{mz_n} \tag{18}$$

for $t$ we have

$$p_t = \frac{e^{z_t}}{S} \tag{19}$$

since $p_i$ is positive for any $i$, we can take the reciprocal of the equation

$$\frac{1}{p_t} = \frac{S}{e^{z_t}} = \frac{e^{z_1} + \cdots + e^{z_t} + \cdots + e^{z_n}}{e^{z_t}} \tag{20}$$

in other words

$$\frac{1}{p_t} = 1 + \sum_{\substack{i=1 \\ i \neq t}}^{n} \frac{e^{z_i}}{e^{z_t}} \tag{21}$$

similarly, for $p'_t$ we have

$$\frac{1}{p'_t} = 1 + \sum_{\substack{i=1 \\ i \neq t}}^{n} (\frac{e^{z_i}}{e^{z_t}})^m \tag{22}$$

Based on (15)

$$\frac{e^{z_i}}{e^{z_t}} < 1, \quad \forall \quad i \neq t \tag{23}$$

which means for $m > 1$

$$\sum_{\substack{i=1 \\ i \neq t}}^{n} \left(\frac{e^{z_i}}{e^{z_t}}\right)^m < \sum_{\substack{i=1 \\ i \neq t}}^{n} \frac{e^{z_i}}{e^{z_t}} \tag{24}$$

$$\frac{1}{p'_t} < \frac{1}{p_t} \tag{25}$$

$$p'_t > p_t \tag{26}$$

On the other hand, for $0 < m < 1$

$$\sum_{\substack{i=1 \\ i \neq t}}^{n} \left(\frac{e^{z_i}}{e^{z_t}}\right)^m > \sum_{\substack{i=1 \\ i \neq t}}^{n} \frac{e^{z_i}}{e^{z_t}} \tag{27}$$

$$p'_t < p_t \tag{28}$$

## D   Ablation Study on RLS Defense Components

The backbone of RLS lies in two key components: the distribution probability used for sampling the coefficient $m$ and the parameters of candidate distributions. We initially focused on the Gaussian and uniform distributions, and found that uniform distribution works surprisingly well, as demonstrated in Section 6.4, outperforming other defenses by a large margin, notably reducing the success rate of the powerful Square attack by at least 40%.

Table 8 summarizes the results of our experiments on the main components of RLS. We run the Square attack (Andriushchenko et al., 2020) against our trained ResNet-18 and use all the 10000 testing samples of CIFAR-10 to attack the model. We implement RLS using Gaussian and uniform distributions. The attacker's query budget is set to 10000 queries, and the detailed settings for the Square attack can be found in Appendix A. It should be noted that since sampling from the Gaussian distribution can result in negative values for $m$, we have set a minimum sampling threshold of 0.2 for all of the experiments conducted using Gaussian distribution.

Table 8: The defense robustness of ResNet-18 against Square Attack under various distributions and parameters for RLS over 10,000 test samples from CIFAR-10. The perturbation size is $\epsilon = 12.75/255$ (or 0.05) and the $\ell_\infty$ norm is used.

| Dist. | Parameters | ASR | Avg. Queries |
|---|---|---|---|
| Uniform | $\mathcal{U}(a = 0.1, b = 10)$ | 57.9% | 78.58 |
| | $\mathcal{U}(a = 0.1, b = 100)$ | 52.7% | 40.69 |
| | $\mathcal{U}(a = 0.5, b = 10)$ | 69.3% | 387.97 |
| | $\mathcal{U}(a = 0.5, b = 100)$ | 60.4% | 21.92 |
| | $\mathcal{U}(a = 0.5, b = 1000)$ | 53.0% | 21.32 |
| Gaussian | $\mathcal{N}(\mu = 1, \sigma = 5)$ | 88.5% | 1211.31 |
| | $\mathcal{N}(\mu = 1, \sigma = 10)$ | 76.9% | 1397.76 |
| | $\mathcal{N}(\mu = 1, \sigma = 100)$ | 49.8% | 41.75 |

The results show that the uniform distribution outperforms the Gaussian distribution in most cases. However, Gaussian distribution's effectiveness against adversarial example attacks is comparable to that of uniform distribution, specifically when $\mu = 1$ and $\sigma = 100$. The results also indicate that for the uniform distribution, the attack success rate drops as the upper-bound $b$ (lower-bound $a$) increases (decreases). For all our evaluations in this work, we employed the uniform distribution for RLS, leaving a more comprehensive investigation into the choice of distribution for future work.

# E   Adaptive Attack Loss Correlation

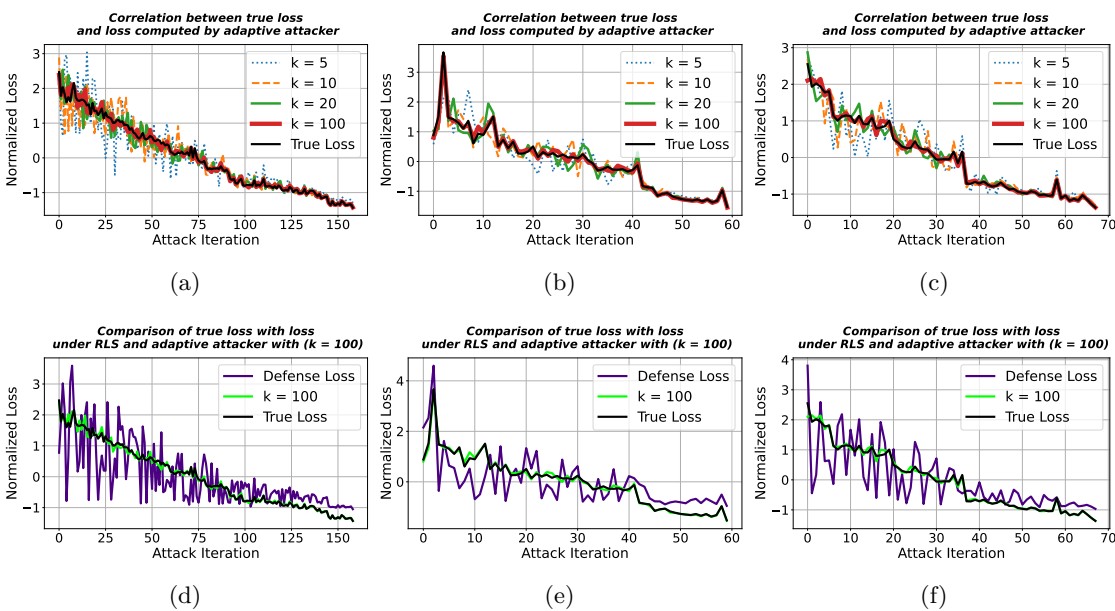

(a)                                   (b)                                   (c)

(d)                                   (e)                                   (f)

Figure 3: The correlation between the loss values under adaptive attacks. All the loss values are normalized to make comparison easier. Three candidate samples from CIFAR-10 are used. (a) through (c) plot the loss values for the undefended model (black) and four configurations of the adaptive attack ($k = 5$, $k = 10$, $k = 20$, and $k = 100$). The plots reveal that as $k$ increases, so does the correlation between the adaptive attack loss values and the true loss values. The plots also show that while the defense loss has large fluctuations and deviations, the adaptive attack's loss for $k = 100$ is closely correlated to the true loss. However, even for $k = 20$, a costly setting for the attacker, adaptive loss values diverge significantly from the actual loss values. (d) to (f) plot the normalized loss values for the undefended model (black), the defended model with no adaptive mechanism (magenta), and with an adaptive mechanism (green).

Figures 3 indicates how increasing $k$ could lead to a better correlation between the true loss and the loss under RLS. Figures 3a, 3b, and 3c demonstrate the correlation between true loss and the loss value computed by an adaptive attacker for four different values of $k$. As the graphs show, increasing $k$ results in more correlation between the loss value computed by the adaptive attack and the true loss value of the undefended model. However, increasing $k$ makes the attack more expensive for an attacker, i.e., the attacker has to use $k$ queries at each iteration of the attack. For instance, when $k = 100$, a budget of 10,000 queries is equivalent to only 500 attack iterations. Figures 3d, 3e, and 3f plot normalized loss values for the true loss (black), loss in the presence of RLS with no adaptive attacker (magenta), and loss in the presence of RLS and an adaptive attacker with $k = 100$ (green). The figures show how a large $k$ could improve the correlation of the loss computed by an attacker with the true loss values.

# F   Additional Experiments

## F.1   Defense Performance against Attacks in $\ell_2$ Space

The experiments in Section 6 are all conducted in $\ell_\infty$ space. Here we provide additional experiments in the $\ell_2$ space and compare the robustness of our defense with other randomized defenses. The results indicate that our defense consistently outperforms other methods in the $\ell_2$ space too. The perturbation budget for experiments in $\ell_2$ is set to 255 (or 1) for CIFAR-10 and 1275 (or 5) for ImageNet. We use the same samples from CIFAR-10 and ImageNet that we used for experiments in the $\ell_\infty$ space (outlined in 6.1) and the same attack and defense settings outlined in Appendix A. We use Bandit (Ilyas et al., 2019), Square (Andriushchenko et al., 2020), and Zo-signSGD (Liu et al., 2019) attacks and victim models are ResNet-18 and VGG-16 for

CIFAR-10, and ResNet-50 for ImageNet. Tables 9, 10, 11 report ASR and average number of queries for ResNet-18, VGG-16, and ResNet-50, respectively.

Table 9: Performance of defense methods against Bandit, Square, and Zo-signSGD attacks with a perturbation budget of 255 ($\ell_2$) evaluated on ResNet-18 using 1,000 random samples from CIFAR-10.

| Defense | Bandit | Square | Zo-signSGD |
|---|---|---|---|
| Undefended | 98.76% / 961.32 | 97.83% / 830.28 | 56.02% / 694.10 |
| iRND ($\nu = 0.01$) | 14.52% / 1502.34 | 32.39% / 2060.24 | 50.69% / 935.96 |
| iRND ($\nu = 0.02$) | 25.66% / 1668.24 | 34.98% / 1460.73 | 41.56% / 1116.96 |
| oRND ($c = 1$) | 17.60% / 2136.53 | 26.29% / 1395.87 | 45.24% / 925.85 |
| RFD ($\sigma = 2.5$) | 24.25% / 2125.35 | 30.81% / 1855.04 | 44.44% / 953.40 |
| RLS (0.5, 10) | **5.92% / 1709.42** | 25.32% / 5189.30 | 36.01% / 867.03 |
| RLS (0.5, 1000) | 6.02% / 1964.12 | **9.65% / 658.78** | **34.86% / 948.09** |

Table 10: Performance of defense methods against Bandit, Square, and Zo-signSGD attacks with a perturbation budget of 255 ($\ell_2$) evaluated on VGG-16 using 1,000 random samples from CIFAR-10.

| Defense | Bandit | Square | Zo-signSGD |
|---|---|---|---|
| Undefended | 98.07% / 1463.61 | 96.23% / 2054.00 | 30.29% / 681.56 |
| iRND ($\nu = 0.01$) | 8.08% / 1657.52 | 29.07% / 3604.49 | 28.85% / 736.11 |
| iRND ($\nu = 0.02$) | 14.57% / 1355.37 | 20.44% / 1922.56 | 24.79% / 1155.64 |
| oRND ($c = 1$) | 11.87% / 2146.18 | 18.07% / 1922.56 | 26.45% / 1119.60 |
| RFD ($\sigma = 0.35$) | 28.02% / 1464.05 | 30.14% / 1488.91 | 29.36% / 1318.08 |
| RLS (0.5, 10) | 3.78% / 1100.05 | 16.60% / 4926.59 | 20.80% / 906.11 |
| RLS (0.5, 1000) | **3.56% / 1347.45** | **7.55% / 1627.91** | **19.40% / 816.35** |

Table 11: Performance of defense methods against Bandit, Square, and Zo-signSGD attacks with a perturbation budget of 1275 (5) ($\ell_2$) evaluated on ResNet-50 using 1,000 random samples from ImageNet.

| Defense | Bandit | Square | Zo-signSGD |
|---|---|---|---|
| Undefended | 99.47% / 2308.49 | 98.52% / 1264.04 | 96.67% / 2069.97 |
| iRND ($\nu = 0.01$) | 73.75% / 6840.36 | 30.08% / 849.83 | 96.22% / 2289.55 |
| iRND ($\nu = 0.02$) | 73.73% / 6492.32 | 27.77% / 1120.32 | 91.48% / 2963.41 |
| oRND ($c = 1$) | 82.87% / 5326.31 | 38.81% / 1214.83 | 83.68% / 2648.10 |
| RFD ($\sigma = 3.0$) | 94.48% / 3310.75 | 68.38% / 915.96 | 83.71% / 2408.99 |
| RLS (0.5, 10) | 67.83% / 7732.39 | 19.11% / 444.94 | 66.76% / 2969.55 |
| RLS (0.5, 1000) | **65.83% / 7865.88** | **12.29% / 179.06** | **66.23% / 3045.12** |

## F.2 Defense Performance against the SOTA Sparse Attack ($\ell_0$)

We evaluate the robustness of RLS and other randomized defenses against BruSLeAttack (Vo et al., 2024), the SOTA sparse black-box score-based attack. In the evaluations, we use ResNet-18 and the same 1,000 randomly selected test samples from CIFAR-10 and also 200 random samples from the 1,000 ImageNet samples that we used in evaluations in Section 6. The hyper-parameter values that we use for BruSLeAttack follow the settings provided by the authors (outlined in Table 7). For evaluations on ResNet-18 (CIFAR-10), we use sparsity levels of 1% and 2.9% corresponding to 10 and 30 pixels, respectively. For evaluations on ResNet-50 (ImageNet), we use sparsity levels of 0.2% and 0.6% which respectively correspond to 100 and 300 pixels. The results are outlined in Tables 12 and 13 and show that RLS outperforms other defenses too.

Table 12: Attack success rate and average number of queries for successfully crafted adversarial examples by BruSLeAttack (Vo et al., 2024) against ResNet-18 (CIFAR-10) with various defensive methods under sparsity levels of 1% and 2.9%.

| Defense | Sparsity of 1% | Sparsity of 2.9% |
|---|---|---|
| Undefended | 99.07% / 98.21 | 100.0% / 21.54 |
| iRND ($\nu = 0.01$) | 98.43% / 108.02 | 100.0% / 25.07 |
| iRND ($\nu = 0.02$) | 96.86% / 168.09 | 99.89% / 31.98 |
| oRND ($c = 1$) | 88.97% / 253.94 | 98.34% / 50.61 |
| RFD ($\sigma = 2.5$) | 87.11% / 272.82 | 97.47% / 96.20 |
| RLS (0.5, 10) | 83.09% / 878.07 | 94.09% / 287.95 |
| RLS (0.5, 1000) | **54.05% / 546.34** | **79.36% / 64.87** |

Table 13: Attack success rate and average number of queries for successfully crafted adversarial examples by BruSLeAttack (Vo et al., 2024) against ResNet-50 (ImageNet) with various defensive methods under sparsity levels of 0.2% and 0.6%.

| Defense | Sparsity of 0.2% | Sparsity of 0.6% |
|---|---|---|
| Undefended | 100.0% / 424.6 | 100.0% / 160.27 |
| iRND ($\nu = 0.01$) | 79.87% / 412.26 | 97.47% / 429.34 |
| iRND ($\nu = 0.02$) | 70.95% / 524.80 | 88.00% / 255.05 |
| oRND ($c = 1$) | 82.76% / 414.02 | 93.01% / 282.11 |
| RFD ($\sigma = 3.0$) | 70.67% / 349.66 | 88.61% / 202.05 |
| RLS (0.5, 10) | 46.84% / 318.23 | 73.42% / 493.06 |
| RLS (0.5, 1000) | **37.34% / 112.93** | **60.76% / 53.17** |

### F.3 Defense Performance against Larger Perturbation Size

The experiments in Section 6 all use a perturbation budget of $\epsilon = 0.05$ or $12.75/255$. Here we evaluate and compare the performance of RLS against other defenses under a larger perturbation budget of $\epsilon = 25.5/255$. The results indicate that RLS consistently outperforms other defenses across a range of perturbation budgets and settings. We employ the Square (Ilyas et al., 2019), attack, Bandit (Ilyas et al., 2019) attack, and SignHunter (Al-Dujaili & O'Reilly, 2020) attack and report ASR and average number of queries. The evaluations are conducted on ResNet-18 and VGG-16 for CIFAR-10, and on ResNet-50 for ImageNet, with the results reported in Tables 15, 16, and 17 respectively.

### F.4 Comparison of robustness between AAA and RLS

In Section 4, we proposed a novel adaptive attack against AAA and experimentally showed AAA's vulnerability to this adaptive attack in Section 6.2. However, for completeness, we provide and compare the results on the defense robustness of AAA-sine and RLS. AAA-linear is excluded from this comparison, as it was demonstrated to be vulnerable to a simple adaptive attack in the original work by Chen et al. (2022). We use the same samples and settings that we used in Tables 4 and 5.

We use Square (Andriushchenko et al., 2020) and SignHunter (Al-Dujaili & O'Reilly, 2020) attacks, the two most efficient attacks based on our results in Tables 4 and 5 and report the results on ImageNet and CIFAR-10 in Table 14. Each cell in the table represents ASR and average number of queries for successfully crafted adversarial examples. The results of Table 14 demonstrate that RLS outperforms AAA-sine across all the models and against bot attacks.

Table 14: The defense robustness of AAA-sine compared with RLS. $\ell_\infty$ with a perturbation budget of $\epsilon = 0.05$ is used.

| Model / Dataset | Defense | Square | SignHunter |
|---|---|---|---|
| ResNet-18 / CIFAR-10 | AAA-sine | 45.80% / 114.51 | 32.61% / 242.61 |
| | RLS (0.5, 1000) | **43.05% / 92.10** | **17.85% / 978.58** |
| VGG-16 / CIFAR-10 | AAA-sine | 27.05% / 108.19 | 31.25% / 264.66 |
| | RLS (0.5, 1000) | **21.56% / 38.58** | **15.85% / 224.44** |
| WideResNet-28-10 / CIFAR-10 | AAA-sine | 46.40% / 120.87 | 34.10% / 436.00 |
| | RLS (0.5, 1000) | **41.38% / 127.47** | **15.08% / 1168.73** |
| ResNet-50 / ImageNet | AAA-sine | 57.20% / 103.66 | 41.00% / 208.26 |
| | RLS (0.5, 1000) | **48.52% / 13.76** | **21.41% / 104.46** |

Table 15: Performance of defense methods against Bandit, Square, and SignHunter attacks with a perturbation budget of 0.1 ($\ell_\infty$) evaluated on ResNet-18 using 1,000 random samples from CIFAR-10.

| Defense | Square | SignHunter | Bandit |
|---|---|---|---|
| Undefended | 100% / 19.52 | 100% / 118.04 | 100% / 61.65 |
| iRND ($\nu = 0.01$) | 99.69% / 52.35 | 97.69% / 274.79 | 90.54% / 319.45 |
| iRND ($\nu = 0.02$) | 97.59% / 68.24 | 89.37% / 446.57 | 90.90% / 194.98 |
| oRND ($c = 1$) | 96.44% / 69.75 | 68.91% / 770.67 | 93.72% / 132.82 |
| RFD ($\sigma = 2.5$) | 96.68% / 127.34 | 67.96% / 786.64 | 96.05% / 196.47 |
| RLS (0.5, 10) | 93.78% / 357.41 | 40.77% / 787.85 | 55.50% / 37.90 |
| RLS (0.5, 1000) | **85.48% / 37.84** | **36.93% / 768.46** | **0.52% / 3.0** |

## F.5 Experiments with Larger $k$ for EOT Adaptive Attack

Table 18 shows the results for larger $k$ values of the EOT adaptive attack using Square Attack against ResNet-18 across our defense method and the baseline defenses. The results show that, although the overall attack success rate increases with larger $k$ values, RLS continues to outperform the baselines, even under the extreme and computationally expensive setting of using $k = 100$ EOT queries per sample.

## F.6 Attack Success Rates Across Query Budgets

Table 19 shows the robustness of RLS and other defenses across query budgets of 1000, 2000, 5000, and 10000. As shown in the table, the attack success rate generally increases with larger query budgets. However, the increase for RLS is substantially smaller than that of the baseline defenses, exhibiting only an approximately 2% increase in ASR from a query budget of 5000 to 10000. These results suggest that RLS provides genuine robustness against the attack rather than simply postponing successful attacks to higher query budgets.

Table 16: Performance of defense methods against Bandit, Square, and SignHunter attacks with a perturbation budget of 0.1 ($\ell_\infty$) evaluated on VGG-16 using 1,000 random samples from CIFAR-10.

| Defense | Square | SignHunter | Bandit |
|---|---|---|---|
| Undefended | 100.00% / 37.55 | 100% / 92.47 | 100% / 85.82 |
| iRND ($\nu = 0.01$) | 99.79% / 48.70 | 99.90% / 117.21 | 62.11% / 678.10 |
| iRND ($\nu = 0.02$) | 98.48% / 93.05 | 98.59% / 181.11 | 61.42% / 392.69 |
| oRND ($c = 1$) | 88.14% / 122.22 | 81.07% / 208.68 | 60.72% / 243.98 |
| RFD ($\sigma = 0.35$) | 89.74% / 323.78 | 84.94% / 483.64 | 74.45% / 318.79 |
| RLS (0.5, 10) | 72.74% / 581.40 | 48.50% / 476.71 | 31.15% / 25.55 |
| RLS (0.5, 1000) | **66.92% / 28.54** | **52.27% / 321.19** | **1.73% / 8.81** |

Table 17: Performance of defense methods against Bandit, Square, and SignHunter attacks with a perturbation budget of 0.1 ($\ell_\infty$) evaluated on ResNet-50 using 1,000 random samples from ImageNet.

| Defense | Square | SignHunter | Bandit |
|---|---|---|---|
| Undefended | 100.00% / 20.47 | 100.00% / 250.52 | 100.00% / 207.23 |
| iRND ($\nu = 0.01$) | 98.52% / 41.78 | 88.32% / 310.61 | 84.31% / 398.40 |
| iRND ($\nu = 0.02$) | 97.28% / 28.65 | 83.74% / 286.15 | 85.29% / 438.90 |
| oRND ($c = 1$) | 97.57% / 39.14 | 80.45% / 430.02 | 91.91% / 327.89 |
| RFD ($\sigma = 3.0$) | 99.42% / 46.86 | 86.76% / 420.74 | 96.41% / 174.39 |
| RLS (0.5, 10) | 93.59% / 84.22 | 42.85% / 337.28 | 86.52% / 324.28 |
| RLS (0.5, 1000) | **86.91% / 27.72** | **39.51% / 146.22** | **29.64% / 23.01** |

Table 18: Defense robustness against the EOT adaptive attack incorporated into the Square attack on ResNet-18 using 1,000 samples from CIFAR-10. The perturbation budget is $\epsilon = 12.75/255$ (or 0.05) and the $\ell_\infty$ norm is used.

| Defense | Parameters | $k = 25$ ASR | $k = 25$ Avg. Q. | $k = 50$ ASR | $k = 50$ Avg. Q. | $k = 100$ ASR | $k = 100$ Avg. Q. |
|---|---|---|---|---|---|---|---|
| RLS | (0.5, 10) | **62.45%** | 974.49 | 72.20% | 1183.94 | 74.07% | 2195.28 |
| | (0.5, 100) | 63.91% | 634.85 | 72.90% | 1316.18 | **71.89%** | 2075.80 |
| | (0.5, 1000) | 75.73% | 889.18 | **70.13%** | 1175.84 | 73.24% | 2040.73 |
| iRND | $\nu = 0.01$ | 96.97% | 1264.30 | 92.79% | 1823.16 | 83.69% | 2321.23 |
| | $\nu = 0.02$ | 94.94% | 1265.97 | 91.38% | 1653.55 | 84.92% | 2181.38 |
| oRND | $c = 1$ | 87.18% | 958.24 | 88.94% | 1479.32 | 84.36% | 1997.14 |
| RFD | $\sigma = 2.5$ | 84.89% | 821.71 | 86.57% | 1319.54 | 83.85% | 1915.30 |

Table 19: Defense robustness results. Entries are reported as ASR (%) / Avg. Queries.

| Defense | Parameter | Budget = 1000 | Budget = 2000 | Budget = 5000 | Budget = 10000 |
|---|---|---|---|---|---|
| RLS | (0.5, 10) | 51.30 / 107.95 | 53.22 / 137.34 | 54.32 / 295.62 | 56.41 / 542.43 |
| | (0.5, 100) | 40.8 / 90.58 | 41.2 / 121.34 | 41.8 / 238.31 | 43.37 / 356.46 |
| | (0.5, 1000) | 44.5 / 57.22 | 44.5 / 57.22 | 44.5 / 57.22 | 46.17 / 89.83 |
| iRND | $\nu = 0.01$ | 76.10 / 163.73 | 79.10 / 196.03 | 81.80 / 252.89 | 86.39 / 378.34 |
| | $\nu = 0.02$ | 65.40 / 142.47 | 67.90 / 209.13 | 68.50 / 369.23 | 73.59 / 552.85 |
| RFD | $\alpha = 2.5$ | 61.70 / 113.49 | 63.70 / 196.06 | 65.70 / 397.38 | 68.95 / 595.08 |
| oRND | $c = 1$ | 62.20 / 107.81 | 63.00 / 141.06 | 64.30 / 280.85 | 67.05 / 420.28 |
| AAA | AAA-Sine | 44.0 / 90.67 | 44.3 / 109.2 | 44.5 / 141.28 | 46.21 / 210.93 |

