# OpenReview forum: "Random Logit Scaling: Defending Deep Neural Networks Against Black-Box Score-Based Adversarial Example Attacks"
_TMLR — Accepted by TMLR_

### Review · Reviewer_GevU · 2026-05-19

**Summary Of Contributions:**

### Summary

This paper proposes a new technique to defend against adversarial examples. In particular, the paper focuses on defending against black-box attacks in which the adversary exploits the model’s classification scores to generate adversarial samples. The main idea is to scale the model’s logit output by a random factor to misguide the attacker’s effort to estimate the gradient direction. The paper also proposes an adaptive attack to show the vulnerability of the existing defense. With a set of experiments, the paper demonstrates the effectiveness of the proposed method.

### Strengths
1. The paper successfully demonstrated the vulnerability of the existing method, effectively showing the motivation for seeking further improvement.
2. The defense performance was measured against many attack methods and many competitors. The result shows significant improvement in the attack success rate and the number of required queries.
3. The paper presents an adaptive attack that estimates the random value from multiple queries. Consideration of such an adaptive attack should be encouraged. The paper also reports experimental results for this attack.

### Weaknesses
1. I don’t see any significant problems or weaknesses in this paper, aside from some minor questions in the **Requested Changes** section.

**Audience:**

Yes

**Audience Explanation:**

To the best of my knowledge, black-box attacks that exploit the model’s score function via zeroth-order optimization are common, so the adversary is likely to use this attack. Defending against such attacks is an important topic in adversarial example research. Also, the paper showed the vulnerability of a recent defense, which implies the need for more advanced defenses.

**Broader Impact Concerns:**

I don’t see a particular broader impact concern regarding this paper.

**Claims And Evidence:**

Yes

**Claims Explanation:**

The main idea is simple and clear to understand. As mentioned above, the paper presents performance results from experiments against many attack methods and competitors. The results are convincing enough to demonstrate the improvement.

**Requested Changes:**

1. While the derived objective functions in Section 5.1 seem reasonable to me, whether or not those objective functions cover all existing attacks (including the attacks used in the experiments) might need more thorough discussion. If the defense works against attacks that use different objective functions, it would be better to provide some justification for why the defense's performance generalizes beyond the context of Section 5.1.
2. Minor writing issues
    * Please arrange Tables and Figures near where they are referred to. For example, Figure 2 is never referred to before the last page of the paper, Tables 2 and 3 should be near Section 6.4, and Table 4 should be near Section 6.5. Mostly, they are one-page before and after the reference, which makes the paper harder to read. It seems feasible to rearrange the Tables and Figures within the given page limit.

---

> ### Author Response · Authors · 2026-06-07
> **Official Comment by Authors**
>
> We appreciate and thank you for your suggestions and comments.
>
> **Q1: While the derived objective functions in Section 5.1 seem reasonable to me, whether or not those objective functions cover all existing attacks (including the attacks used in the experiments) might need more thorough discussion. If the defense works against attacks that use different objective functions, it would be better to provide some justification for why the defense's performance generalizes beyond the context of Section 5.1.**
>
> We agree that the connection between Section 5.1 and the evaluated attacks should be made clearer. The objectives in Section 5.1 are intended to show representative margin loss objectives for targeted and untargeted score-based attacks, where score-based attacks may use the exact objective (such as Square Attack [4]) or a surrogate loss for the same objective. We will expand Section 5.1 to address this. In particular, we will revise and expand the final paragraph of Section 5.1 to state that “the objective functions in Eqs. (1)–(4) are representative margin loss objectives rather than an exhaustive description of all score-based attacks” and that “score-based attacks may use surrogate losses or alternative optimization procedures that essentially pursue the same goal.” We will further clarify that "RLS generalizes across score-based attacks because it corrupts the score information on which both exact and surrogate objectives rely."
>
> We clarify that RLS generalizes across score-based objective functions used by adversaries by misguiding both gradient-estimation and search-based methods used by these attacks, as explained in Section 3.2.1. To be more specific, **by randomizing $m$ at every query, finite-difference calculations become misguided; hence, the true directional gradient is masked by the noise introduced by the random scaling (affecting attacks such as NES [1], Bandit [2], ZO-signSGD [3]). Furthermore, search-based methods also become unreliable due to the random fluctuations of $m$, which lead to falsified scores and therefore incorrect decisions during the search (affecting Square Attack [4] and SignHunter [5]).**
>
>
>
> **Q2: Minor writing issues: Please arrange Tables and Figures near where they are referred to. For example, Figure 2 is never referred to before the last page of the paper, Tables 2 and 3 should be near Section 6.4, and Table 4 should be near Section 6.5. Mostly, they are one-page before and after the reference, which makes the paper harder to read. It seems feasible to rearrange the Tables and Figures within the given page limit.**
>
> **R2**: We agree with and appreciate the proposed rearrangement of the tables and figures. We will improve the placement of the figures and tables in the revied version according to the reviewer's suggestion.
>
> ---
> [1] Andrew Ilyas, Logan Engstrom, Anish Athalye, and Jessy Lin. "Black-box adversarial attacks with limited queries and information". In 35th International Conference on Machine Learning, ICML, 2018.
>
> [2] Andrew Ilyas, Logan Engstrom, and Aleksander Madry. "Prior convictions: Black-box adversarial attacks with bandits and priors". In The 7th International Conference on Learning Representations, ICLR, 2019
>
> [3] Sijia Liu, Pin-Yu Chen, Xiangyi Chen, and Mingyi Hong. "SignSGD via zeroth-order oracle". In The 7th International Conference on Learning Representations, ICLR, 2019.
>
> [4] Maksym Andriushchenko, Francesco Croce, Nicolas Flammarion, and Matthias Hein. "Square attack: A query-efficient black-box adversarial attack via random search". In 14th European Conference on Computer
> Vision, ECCV, 2020
>
> [5] Abdullah Al-Dujaili and Una-May O’Reilly. "Sign bits are all you need for black-box attacks". In The 8th International Conference on Learning Representations, ICLR, 2020.

---

### Review · Reviewer_vTqN · 2026-05-22

**Summary Of Contributions:**

The article proposes random logit scaling -- a defence against black-box score based adversarial attacks. The idea is simple : Select $m \sim U[a,b]$ and distort the probabilities by multiplication of logits with $m$. The authors also propose a attach against AAA-defence.

Strengths: The idea is simple, and experiments (also follows from design) show that accuracy is preserved while improving the defence. The method does not require re-training and hence is plug-and-play.

Weaknesses: While the claims made in the article are true, there are some flaws with the proposed approach: The proposed method does not help against methods like FGSM (sec 3.1.1). Two contributions seem unrelated and I am not sure why they should have appeared in the same article.

**Additional Comments:**

NA

**Audience:**

Yes

**Audience Explanation:**

Yes. The topic of the article is of interest to several members to the TMLR community.

**Claims And Evidence:**

Yes

**Claims Explanation:**

The key issue is the following : Preserving the ranking by using positive $m$ is presented as an advantage. However, for an attacker the key information they need is only the ordering of the classes which they can exploit. (For instance, for FGSM and the like). So, I think it should be stated that *The proposed method does not work for attacks other than score-based ones**.

Also, even for the score-based approaches one can use some tricks to remove the $m$ dependencies. For instance, take $\log p'_i = mz_i - \log S'$, then subtract the mean across classes to remove the $-\log S'$ constant. This recovers a vector proportional to $mz_i$, which has the same gradient direction as $z_i$. Since $m > 0$ is the same scalar applied to all logits, gradient direction is fully recovered.

I do agree with the claim that, accuracy drop is $0$. However, this might be misleading since several attacks are not addressed.

I also could not find any issues with the attack against AAA.

**Requested Changes:**

Please address the issues raised above. Apart from that certain moderations of the claims and rewriting of text such as  - rank preservation of logit scaling is free advantage, evaluation against other forms of attacks and not just score-based -- needs to be done.

---

> ### Author Response · Authors · 2026-06-07
> **Official Comment by Authors**
>
> Thank you for your comments and feedback. We hope that the following responses address your concerns.
>
> **Q1: Preserving the ranking by using positive m is presented as an advantage. However, for an attacker the key information they need is only the ordering of the classes which they can exploit. (For instance, for FGSM and the like). So, I think it should be stated that The proposed method does not work for attacks other than score-based ones.**
>
> **R1:** We agree that decision-based methods rely only on final predictions and not on confidence scores, and that it should be stated in the paper that the proposed method does not work against decision-based attacks. In the initially submitted version, we explicitly mentioned that the focus of our method, following the state-of-the-art AAA defense, which also solely focuses on score-based attacks, is on score-based attacks. This was stated in the title of the paper, as well as in the abstract and introduction. We agree that this point should be made clearer to the reader, and therefore **we will modify Section 5 to ensure that it is reiterated there as well. Additionally, we will add a Discussion Section to the revised version to discuss this limitation and other limitations of RLS in more details.**
>
> The rationale for focusing on score-based attacks, both in our method and in other state-of-the-art defenses such as AAA (which we showed to be vulnerable to a simple adaptive attack) is that decision-based attacks are considerably less practical than score-based attacks. State-of-the-art decision-based attacks typically require hundreds of queries to craft a single adversarial example, compared to only dozens of queries for score-based attacks. **For instance, RayS [1], one of the most powerful decision-based attacks, requires an average of around 800 queries, whereas Square Attack [2], a powerful score-based attack, requires fewer than 100 queries on average while achieving success rates close to 100%.**
>
>
> **Q2: Also, even for the score-based approaches one can use some tricks to remove the m dependencies. For instance, take log p'_i = mz_i - log S', then subtract the mean across classes to remove the -logS' constant. This recovers a vector proportional to mz_i, which has the same gradient direction as z_i. Since m > 0 is the same scalar applied to all logits, gradient direction is fully recovered.**
>
>
> **R2:** We agree that mean-centering the log probabilities removes the $S^\prime$ constant and removes the scalar offset per query. However, it does not remove the random scaling across different queries, which is the primary reason for RLS's success against score-based attacks, as these attacks rely on comparing the outputs of multiple queries to estimate gradients. Specifically, assuming $m_1$ and $m_2$ are selected for two queries $x^{(1)}$ and $x^{(2)}$ with logit vectors $z^{(1)}$ and $z^{(2)}$, the attacker has
>
> $$
> \log{p_i^\prime(x^{(1)})} - \frac{1}{C} \sum_j \log{p_j^\prime(x^{(1)})} = m_1(z^{(1)}_i-\bar{z}^{(1)})
> $$
>
> and similarly, $m_2(z^{(2)}_i-\bar{z}^{(2)})$, which are proportional to $m_1z^{(1)}$ and $m_2z^{(2)}$, respectively. However, the attacker still relies on the difference between the two outputs, which is not proportional to the true loss difference unless $m_2 = m_1$.
>
> Therefore, the aforementioned mean-centering approach only removes the scalar offset per query, while the random scaling factor $m$ remains and continues to prevent the attacker from reliably comparing query outputs.
>
>
> **Q3: I do agree with the claim that, accuracy drop is 0. However, this might be misleading since several attacks are not addressed.**
>
> **R3:** Please refer to our response to **Q1**. As mentioned above in **R1**, following AAA, the proposed defense does not address decision-based attacks due to their impracticality. Therefore, the fact that the accuracy drop is 0 reflects a utility property of RLS under the stated score-based black-box threat model.
>
> Moreover, RLS only rescales the logits and does not alter their relative ordering. Consequently, the predicted class remains unchanged, and the clean accuracy is theoretically preserved regardless of whether decision-based attacks are considered in the evaluation. Thus, the reported 0 accuracy drop is an inherent property of RLS rather than an artifact of excluding specific attack classes.
>
>
> **Q4: I also could not find any issues with the attack against AAA.**
>
> **R4:** We hope that we have understood your point correctly and that our proposed adaptive attack against AAA adequately addresses your concerns.
>
> ---
>
> [1] Chen, Jinghui, and Quanquan Gu. "Rays: A ray searching method for hard-label adversarial attack." Proceedings of the 26th ACM SIGKDD International Conference on Knowledge Discovery & Data Mining. 2020.
>
> [2] Lin, Maximilian, et al. "Square attack: A query-efficient black-box adversarial attack via random search." Proceedings of the European Conference on Computer Vision (ECCV). 2020.

---

### Review · Reviewer_ri5V · 2026-05-29

**Summary Of Contributions:**

This paper proposes Random Logit Scaling (RLS), a simple post-processing defense against black-box score-based adversarial attacks. RLS randomly scales model logits with a positive value, preserving the predicted label while distorting the returned confidence scores. This makes gradient estimation and search-direction evaluation harder for attackers.
The paper also introduces Pendulum, an adaptive attack showing that the prior deterministic defense AAA-sine is vulnerable. Experiments on CIFAR-10 and ImageNet with various models and attacks show that RLS lowers attack success rates while maintaining clean accuracy, outperforming several randomized baselines under limited adaptive attacks.

**Audience:**

Yes

**Audience Explanation:**

This topic is relevant to ML conference readers, especially those interested in adversarial robustness, ML security, trustworthy AI, and practical deployment. The paper addresses an important problem: deployed ML services often expose confidence scores that can be exploited by black-box adversarial attacks. RLS is appealing because it is simple, lightweight, and can be applied without retraining or modifying the model.

**Claims And Evidence:**

No

**Claims Explanation:**

The paper provides solid empirical support that RLS reduces the effectiveness of standard black-box score-based attacks while preserving top-1 accuracy. Its core idea is mathematically sound because positive logit scaling does not change the predicted class. Experiments across datasets, models, attacks, and perturbation norms show that RLS often lowers attack success rates compared to other randomized defenses, while avoiding clean accuracy loss.

However, the evidence is not strong enough to fully prove that RLS is a state-of-the-art defense against well-resourced adaptive attackers. Since RLS relies on randomness, stronger EOT-based evaluations with larger averaging budgets are needed. The paper also lacks broader robustness metrics, such as robust accuracy over query budgets and failure-inclusive query statistics. In addition, while RLS reduces L2 score distortion, its calibration results are mixed. Overall, RLS is promising under the tested threat model, but its robustness against stronger adaptive attacks remains uncertain.

**Requested Changes:**

To improve the paper’s chances of acceptance, I recommend several revisions.

1.  the adaptive attack evaluation should be strengthened. Because RLS is a randomized defense, attackers may naturally average multiple outputs from the same or nearby inputs. The paper should therefore evaluate larger EOT or averaging budgets, such as k = 20, 50, 100, or higher, and report how attack success changes as the total query budget increases.

2.  robustness should be reported as a function of query budget. Curves showing attack success rate or robust accuracy over the number of queries would clarify whether RLS genuinely improves robustness or mainly delays successful attacks.

3.  the threat model needs clarification. The paper should state whether attackers can repeatedly query the same input, whether API limits exist, whether attackers know the scaling-factor distribution, and whether they know the defense is deployed.

4.  the paper should discuss more seriously the trade-off between robustness and confidence quality. Although RLS preserves top-1 accuracy, it may distort confidence scores and worsen calibration, which matters for systems relying on confidence estimates in rejection, human review, OOD detection, or medical decision-making.

5. the authors should provide a more principled justification for the scaling distribution and parameter choices, including how different scaling ranges affect attack success, calibration error, numerical stability, and confidence distortion. The comparison with baselines should also be more transparent, with fair tuning procedures and sensitivity analyses for key hyperparameters.

6.  the paper should moderate its claims. RLS appears promising and practical under the evaluated settings, but it should not be framed as a definitive state-of-the-art defense without stronger adaptive evaluations.

---

> ### Author Response · Authors · 2026-06-09
> **Official Comment by Authors (1/3)**
>
> We appreciate the detailed feedback and comments. We hope to address your concerns in the following.
>
> **Q1: the adaptive attack evaluation should be strengthened. Because RLS is a randomized defense, attackers may naturally average multiple outputs from the same or nearby inputs. The paper should therefore evaluate larger EOT or averaging budgets, such as k = 20, 50, 100, or higher, and report how attack success changes as the total query budget increases.**
>
>
> **R1:** We agree that for randomized defenses, adaptive attackers can reduce the effect of randomization defense by averaging multiple outputs through EOT. This limitation is shared by prior randomized defenses such as iRND, oRND, and RFD as well. Therefore, following the related work, we evaluated RLS against this adaptive attack using EOT and compared the results for $k=1$, $k=5$, and $k=10$ in Section 6.5.
>
> Here we provide the results for $k=25$, $k=50$, and $k=100$ as well. Each cell represents the attack success rate and average number of queries used for successful evasive samples. As expected, attack success rates generally increases as $k$ grows because the attacker obtains a more accurate estimate of the expected model output. However, even under this stronger adaptive attack, RLS continues to outperform all competing defenses in terms of attack success rate. We omitted AAA since we showed that even without considering EOT attacks, AAA is susceptible to a simple heuristic-based adaptive attacker.
>
> | Defense | Parameter | $k=25$ | $k=50$ | $k=100$ |
> | :---: | :---: | :---: | :---: | :---: |
> | **RLS** | (0.5, 10) | 62.45 / 974.49 | 72.20 / 1183.94 | 74.07 / 2195.28 |
> | | (0.5, 100) | 63.91 / 634.85 | 72.90 / 1316.18 | 71.89 / 2075.80 |
> | | (0.5, 1000) | 75.73 / 889.18 | 70.13 / 1175.84 | 73.24 / 2040.73 |
> | **iRND** | $v$= 0.01 | 96.97 / 1264.30 | 92.79 / 1823.16 | 83.69 / 2321.23 |
> | | $v$ = 0.02 | 94.94 / 1265.97 | 91.38 / 1653.55 | 84.92 / 2181.38 |
> | **RFD** | $c$ = 1 | 84.89 / 821.71 | 86.57 / 1319.54 | 83.85 / 1915.30 |
> | **oRND** | $\alpha$ = 2.5 | 87.18 / 958.24 | 88.94 / 1479.32 | 84.36 / 1997.14 |
> | | | | ||
>
> We would also like to note that one of the main objectives of randomized defenses is not to provide certified robustness, but rather to increase the attacker's cost by requiring substantially more queries to obtain reliable estimates. While larger EOT budgets can reduce the effect of randomization, they also increase the attacker's query complexity, making attacks more expensive and potentially easier to detect in practical deployments.
>
> **We will add the results of $k=50$ and $k=100$ to Table 4. Additinoally, we will add a "Discussion" Section to the paper to mention the limitations of RLS including its randomized nature which does not provide any certified robustness and mainly increases the cost for an adaptive attacker.**
>
>
> **Q2: Robustness should be reported as a function of query budget. Curves showing attack success rate or robust accuracy over the number of queries would clarify whether RLS genuinely improves robustness or mainly delays successful attacks.**
>
> **R2:** Here we provide attack success rate for query budgets of 1000, 2000, and 5000. Additionally, in all of our reported results, following the standard numbers used in the related work, we used a budget of 10,000 queries and showed that RLS decreses the **success rate** of all the considered attacks significantly more than that of other defenses. This shows that RLS effectively improves robustness rather than just delaying successful attacks. **We will add these results in the Experiments section of the revision as well.**
>
> | Defense | Parameter | Budget = 1000 | Budget = 2000 | Budget = 5000 | Budget = 10000 |
> | :---: | :---: | :---: | :---: | :---: | :---: |
> | **RLS** | (0.5, 10) | 51.30 / 107.95 | 53.22 / 137.34 | 54.32 / 295.62 | 56.41 / 542.43 |
> |  | (0.5, 100) | 40.8 / 90.58 | 41.2 / 121.34 | 41.8 / 238.31 | 43.37 / 356.46 |
> |  | (0.5, 1000) | 44.5 / 57.22 | 44.5 / 57.22 | 44.5 / 57.22 | 46.17 / 89.83 |
> | **iRND** | $v$ =  01 | 76.10 / 163.73 | 79.10 / 196.03 | 81.80 / 252.89 | 86.39 / 378.34 |
> | | $v$ = 0.02 | 65.40 / 142.47 | 67.90 / 209.13 | 68.50 / 369.23 | 73.59 / 552.85 |
> | **RFD** | $c$ = 1 | 61.70 / 113.49 | 63.70 / 196.06 | 65.70 / 397.38 | 68.95 / 595.08 |
> | **oRND** | $\alpha$ = 2.5 | 62.20 / 107.81 | 63.00 / 141.06 | 64.30 / 280.85 | 67.05 / 420.28 |
> | **AAA** | AAA-Sine  |  44.0 / 90.67 |  44.3 / 109.2  | 44.5 / 141.28 | 46.21 / 210.93 |
> |  | |  |  |  | |

---

> ### Author Response · Authors · 2026-06-09
> **Official Comment by Authors (2/3)**
>
> **Q3: the threat model needs clarification. The paper should state whether attackers can repeatedly query the same input, whether API limits exist, whether attackers know the scaling-factor distribution, and whether they know the defense is deployed.**
>
> **R3:** We agree that the threat model should be made clearer. We will add a subsection at the end of Section  5 to provide all the requested information in a concise manner. To address the reviewer's concern, we outline our threat model here: we consider a black-box attacker who can repeatedly query the same input without API limitations, subject only to a budget of 10,000 queries (which we used across all evaluations). Additionally, we assume the attacker is aware of the deployed defense but that the scaling-factor is not known.
>
>
> **Q4: the paper should discuss more seriously the trade-off between robustness and confidence quality. Although RLS preserves top-1 accuracy, it may distort confidence scores and worsen calibration, which matters for systems relying on confidence estimates in rejection, human review, OOD detection, or medical decision-making.**
>
> **R4:**  We agree that the trade-off between robustness and potential distortion to confidence scores deserves a clearer and more explicit discussion. Currently, we analyze how RLS affects confidence scores and calibration error in Section 6.3 and Table 5. Our results show that while RLS preserves top-1 accuracy by construction, while also preserves the class ranking/order, which can be important in medical applications where other randomized defenses may alter this order. However, it introduces a mild calibration degradation due to the randomized scaling of logits. For instance, on ResNet-18, the Expected Calibration Error (ECE) increases from 0.0268 for the undefended model to 0.0401 for RLS (0.5, 10) and 0.0448 for RLS (0.5, 1000). However, the experiments also demonstrate that RLS causes a lower level of average $L_2$ confidence-score distortion compared to alternative defenses like iRND, oRND, and RFD, while achieving substantially stronger robustness against score-based black-box attacks.
>
> **We will expand the discussion in Section 6.3 to make this trade-off explicit** and clarify that RLS is most appropriate in settings where preserving top-1 predictions and improving robustness are prioritized, whereas applications that rely heavily on calibrated confidence scores may require additional calibration or a carefully chosen distribution for the scaling factor.
>
>
> **Q5-A: the authors should provide a more principled justification for the scaling distribution and parameter choices, including how different scaling ranges affect attack success, calibration error, numerical stability, and confidence distortion.**
>
> **R5-A:** We agree that the choice of scaling distribution and its parameters should be sufficiently motivated. We refer the reviwer to Appendix D where we have conducted an ablation study on the performance of distribution probabilities and their parameters. We conducted an ablation study over multiple distribution choices and parameter settings. Specifically, we evaluated eight different RLS configurations and analyzed their impact on attack success rate. Additionally, to further characterize the robustness and confidence score distortion trade-off, Section 6.3 reports calibration error and confidence-score distortion for the three most robust configurations identified in the ablation study. The results demonstrate that increasing the strength of the random scaling generally improves robustness while also increasing confidence distortion.
>
>
> **Q5-B: The comparison with baselines should also be more transparent, with fair tuning procedures and sensitivity analyses for key hyperparameters.**
>
> **R5-B:** We absolutely agree that the the tuning process for baseline methods needs careful attention. As described in Appendix A, in order to do a fair evaluation and comparison with the baselines, we used the exact best-performing hyperparameter values reported in the original papers for each baseline whenever those values were available. RFD is the only exception to this rule since the original paper did not report the exact hyperparameter values but only the drop in the clean accuracy of the model caused by the defense. Therefore, for RFD we performed a hyperparameter search for each model and selected the configuration that achieved the strongest robustness while maintaining the clean-accuracy value reported in the original work. **Essentially, for all the baselines, we used the authors' reported best settings and therefore did not repeat their hyperparameter sensitivity analyses, except for RFD where the exact hyperparameter values were not reported.**

---

> ### Author Response · Authors · 2026-06-09
> **Official Comment by Authors (3/3)**
>
> **Q6: the paper should moderate its claims. RLS appears promising and practical under the evaluated settings, but it should not be framed as a definitive state-of-the-art defense without stronger adaptive evaluations.**
>
> **R6:** We thank the reviewer and agree that the claims should be moderated and that the claimed security guarantees are not definitive. We will revise the paper to first clearly outline the threat model and secondly to replace the SOTA claim with **"improvement over the baselines" while mentioning that it significantly increases the costs for considered adaptive attacker scenarios**.

---

### Author Response · Authors · 2026-06-12
**Official Comment by Authors: Revision Summary**

We thank all the reviewers for their comments and feedback. To address the reviewers' concerns, we made several changes to our submission and have uploaded the revised version. Here, we outline the changes we made to the manuscript:

### Reviewer GevU

- We added a paragraph at the end of Section 5.1, page 6, to clarify the connection between the objective functions in Eq. (2) and Eq. (5) and score-based attacks, and explained how our approach achieves effectiveness against these attacks.
- We moved Figure 2 to page 11 and rearranged the placement of the tables so that they appear exactly where they are discussed in the text.

### Reviewer vTqN

- At the beginning of Section 5, page 6, we added a sentence reiterating that RLS only perturbs confidence scores and therefore has limited effectiveness against decision-based attacks.
- We expanded the discussion of the limited effectiveness of RLS against decision-based attacks by adding a new section, Discussion (Section 7, page 12).

### Reviewer ri5V
- We added the results for EOT with $k=25$, $k=50$, and $k=100$ to our submission. To adhere to the page limit, we included the results and their discussion in the Appendix as Section F.5, page 22, and added an appropriate reference to this section in the main text (Section 6.5, page 11).
- We also added a Discussion section (Section 7, page 12), in which we outline the limitations of RLS, including its randomized nature and the potential for adaptive attackers to exploit this randomness.
- We added results for attack success rates across multiple query budgets, including budgets of 1000, 2000, 5000, and 10000. To adhere to the page limit, we included these results in the Appendix as Section F.6, page 22, and added appropriate references to them in the main text (Section 6.4, page 10).
- We added a new subsection (Section 5.3, page 8) to the manuscript, providing a more detailed description of the considered threat model.
- We added a paragraph at the end of Section 6.3, page 10, to explicitly describe the trade-off between the robustness provided by RLS and confidence score preservation.
- We revised the wording of the second contribution in the Introduction, page 2, and the last sentence of the Abstract to moderate our state-of-the-art claims.

---

### Decision · Action_Editor_BUGi · 2026-06-22

**Recommendation:** Accept as is

**Audience:**

Yes

**Audience Explanation:**

All three reviewers gave a "yes" on this question.  The paper is definitely of interest to some members of the TMLR audience.

**Claims And Evidence:**

Yes

**Claims Explanation:**

Initially, two of the three reviewers said "yes" to this, and one said "no."  Reviewer ri5V said no, and asked for several additional results, clarifications, etc.  The authors provided these in the rebuttal (and an update to the paper); ultimately, this reviewer indicated that they were happy with the changes and voted to recommend acceptance.

The other two reviewers said "yes" to this question, but had various minor requested changes.  The authors also addressed these adequately in their revision, and again, both of the reviewers advocated for accepting the paper.